# GEO-Bench-2: From Performance to Capability, Rethinking Evaluation in Geospatial AI

**Naomi Simumba**[†]                                    *naomi.simumba@ibm.com*
*IBM Research Europe*

**Nils Lehmann**[†]                                        *n.lehmann@tum.de*
*Technical University Munich, Munich Center for Machine Learning (MCML)*

**Paolo Fraccaro**[†,‡]                                   *paolo.fraccaro@ibm.com*
*IBM Research Europe*

**Hamed Alemohammad**                                 *halemohammad@clarku.edu*
*Clark University*

**Geeth De Mel**                                          *geeth.demel@ibm.com*
*IBM Research Europe*

**Salman Khan**                                       *salman.khan@mbzuai.ac.ae*
*MBZUAI*

**Manil Maskey**                                        *manil.maskey@nasa.gov*
*NASA Impact*

**Nicolas Longepe**                                    *nicolas.longepe@esa.int*
*ESA Φ-lab*

**Xiao Xiang Zhu**                                      *xiaoxiang.zhu@tum.de*
*Technical University Munich, Munich Center for Machine Learning (MCML)*

**Hannah Kerner**                                          *hkerner@asu.edu*
*Arizona State University*

**Juan Bernabe-Moreno**                                  *juan.bernabe@ibm.com*
*IBM Research Europe*

**Alexandre Lacoste**[‡]                         *alexandre.lacoste@servicenow.com*
*ServiceNow Research*

**Reviewed on OpenReview:** *https://openreview.net/forum?id=NPf175jnP1&noteId=us3spKavzp*

## Abstract

Geospatial Foundation Models (GeoFMs) are transforming Earth Observation (EO), but evaluation lacks standardized protocols. GEO-Bench-2 addresses this with a comprehensive framework spanning classification, segmentation, regression, object detection, and instance segmentation across 19 permissively-licensed datasets. We introduce *capability* groups to rank models on datasets that share common characteristics (e.g., resolution, spectral bands,

---

[†] Equal Contribution (Simumba, Lehmann, Fraccaro).
[‡] Corresponding author (Lacoste, Fraccaro).

temporality), enabling users to identify which models excel in each capability and to determine where future work should focus. To support both fair comparison and methodological innovation, we define a prescriptive yet flexible evaluation protocol. This ensures consistency in benchmarking while facilitating research into model adaptation strategies, a key open challenge in advancing GeoFMs for downstream tasks. Our experiments show that no single model dominates across all tasks, confirming that architectural and pre-training choices lead to task-specific strengths. While models pretrained on natural images (ConvNext-ImageNet, DINOv3) excel on high-resolution tasks, EO-specific models (Terra-Mind, Prithvi, and Clay) outperform them on multispectral applications such as agriculture and disaster response. These findings demonstrate that optimal model choice depends on task requirements, data modalities, and operational constraints, and that the goal of a single GeoFM that performs well across all tasks remains open for future research. GEO-Bench-2 enables informed, reproducible GeoFM evaluation tailored to specific use cases. Code, data, and the leaderboard are publicly released under a permissive license (Project page: https://the-ai-alliance.github.io/GEO-Bench-2/).

## 1 Introduction

Deep learning and self-supervision have transformed Earth Observation (EO), enabling analysis of vast, heterogeneous geospatial data at unprecedented scales and accuracy. Geospatial Foundation Models (GeoFMs) are large-scale architectures pre-trained on diverse data that promise to generalize across tasks, sensors, and geographies while reducing reliance on task-specific supervision. Despite rapid progress in multimodal large language models (MLLMs), recent evidence shows that even frontier models continue to lag behind vision specific models on EO classification tasks as well as complex spatial and geospatial reasoning tasks central to EO applications (Cai et al., 2025; Mallya et al., 2025; Danish et al., 2025). Consequently, GeoFMs, due to their domain-specific pretraining, adaptability, and substantially smaller model size (often 10 to 100× fewer parameters), remain a strong and practical choice for many EO workloads. However, their development remains hampered by data complexity and the absence of standardized evaluation protocols. While several EO benchmarks have been proposed, they often focus on limited modalities, tasks, or regions, or rely on aggregated evaluation protocols that obscure differences in model capabilities. This fragmentation makes it difficult to compare models and assess generalization (see Section 2).

We present GEO-Bench-2, a comprehensive framework for evaluating GeoFMs across real-world EO applications. Our contributions include: (1) 19 curated datasets with permissive licenses organized into overlapping capability-specific subsets (pixel-wise, detection, multi-temporal, multi-spectral, etc.); (2) geographically balanced splits for scalable, representative evaluation; (3) a prescriptive but flexible protocol that ensures fair comparison while allowing for innovation in model fine-tuning and adaptation; (4) full integration with TerraTorch (Gomes et al., 2025), lowering barriers to entry; and (5) an interactive leaderboard on Hugging Face for community engagement. GEO-Bench-2 enables targeted capability assessment and reproducible comparison, accelerating progress toward general-purpose Geospatial Foundation Models.

## 2 Related Work

Benchmarking has become a central component in advancing Earth Observation (EO) machine learning, particularly with the rise of Geospatial Foundation Models (GeoFMs). Several recent efforts have introduced large-scale datasets and evaluation suites to support model development across diverse EO tasks.

PANGAEA (Marsocci et al., 2024) and Copernicus-Bench (Wang et al., 2025) provide curated datasets targeting multi-modal and multi-sensor learning, while other initiatives such as FoMo (Bountos et al., 2025), REO-Bench (Li et al., 2025), and PhilEO (Fibaek et al., 2024) focus on specific challenges including representation learning, and regional generalization. These benchmarks have contributed valuable resources for training and evaluating EO models, but often emphasize particular modalities, geographic regions, or task formulations, and typically report performance through aggregate metrics across datasets.

| | Core | Pixel-wise | Classification | Detection | Multi Temporal | < 10m | >=10m | RGB/NIR | Multi-Spectral |
|---|---|---|---|---|---|---|---|---|---|
| Clay-V1 ViT-B | 1 | 1 | 9 | 2 | 2 | 6 | 7 | 4 | 3 |
| ConvNext-XLarge ImageNet | 2 | 3 | 4 | 1 | 8 | 3 | 4 | 2 | 5 |
| DinoV3-ConvNext Large-WEB | 3 | 6 | 1 | 5 | 12 | 2 | 5 | 3 | 4 |
| ConvNext-Large ImageNet | 4 | 4 | 3 | 3 | 10 | 4 | 2 | 5 | 6 |
| Prithvi-EO-2.0-600M-TL | 5 | 5 | 8 | 9 | 4 | 8 | 6 | 8 | 2 |
| TerraMind-V1 Large | 6 | 2 | 6 | 8 | 1 | 7 | 1 | 7 | 1 |
| Satlas-SwinB Sentinel2 | 7 | 7 | 7 | 4 | 6 | 10 | 3 | 9 | 7 |
| DOFA-ViT-L | 8 | 9 | 5 | 12 | 7 | 5 | 8 | 6 | 8 |
| Prithvi-EO-2.0-300M-TL | 9 | 10 | 11 | 6 | 5 | 11 | 11 | 11 | 9 |
| DinoV3-ViT-L SAT | 10 | 8 | 2 | 11 | 9 | 1 | 9 | 1 | 11 |
| TerraMind-V1 Base | 11 | 12 | 12 | 10 | 3 | 13 | 10 | 12 | 10 |
| Satlas-SwinB Naip | 12 | 11 | 10 | 7 | 11 | 9 | 12 | 10 | 12 |
| DeCUR-Resnet50 | 13 | 14 | 13 | 14 | 13 | 14 | 13 | 14 | 13 |
| Resnet50-ImageNet | 14 | 13 | 14 | 13 | 14 | 12 | 14 | 13 | 14 |

Table 1: Model ranking across GEO-Bench-2 capabilities. Lighter color indicates higher positional ranking hence better performance in a capability.

Despite this progress, several limitations remain. Many benchmarks are constrained by restrictive data licensing, limiting reproducibility and broader adoption. Others focus narrowly on specific task types (e.g., classification or segmentation), lacking support for detection or instance-level reasoning. In addition, evaluation protocols are often underspecified or overly rigid, making fair comparison difficult or limiting methodological exploration. GEO-Bench (Lacoste et al., 2023) introduced more structured evaluation practices, but remains limited in task diversity, integration with modern fine-tuning frameworks, and support for analyzing model capabilities beyond aggregate performance metrics.

PANGAEA (Marsocci et al., 2024) represents one of the most comprehensive recent efforts toward standardized evaluation of GeoFMs, providing a diverse set of datasets and a carefully controlled benchmarking protocol. However, its design primarily follows a dataset-centric evaluation paradigm, with a strong emphasis on controlled comparison settings and frozen representations. In addition, it excludes task categories such as object detection and instance segmentation, and reports performance largely through aggregated metrics across datasets.

Motivated by the limitations of frontier multimodal models on complex spatial and geospatial reasoning tasks highlighted in the introduction, GEOBench-VLM (Danish et al., 2025) provides one of the first systematic evaluations of vision-language models (VLMs) and multimodal large language models (MLLMs) in Earth observation settings. GEOBench-VLM spans 31 fine-grained tasks, including object counting, localization, segmentation, temporal reasoning, and non-optical understanding, capabilities that are central to many real-world EO applications. Its evaluations show that, despite strong performance on natural-image benchmarks, state-of-the-art VLMs achieve limited accuracy on geospatial tasks and exhibit consistent failure modes, particularly for small-object reasoning, spatial precision, and temporal change understanding. These findings provide empirical support for the observation that general multimodal reasoning does not yet translate into robust geospatial capability, reinforcing the need for EO-specific models and evaluation protocols that explicitly target such challenges.

## 3 GEO-Bench-2

### 3.1 Dataset Selection and Transformation

There are hundreds of EO labeled datasets (Schmitt et al., 2023). Through an in-depth investigation, we identified 35 datasets with the potential to meet the criteria below; after experimentation and thorough analysis, we selected 19 that enable high-quality GeoFM evaluation (see Table 2 and full details in Appendix C). Our selection criteria were:

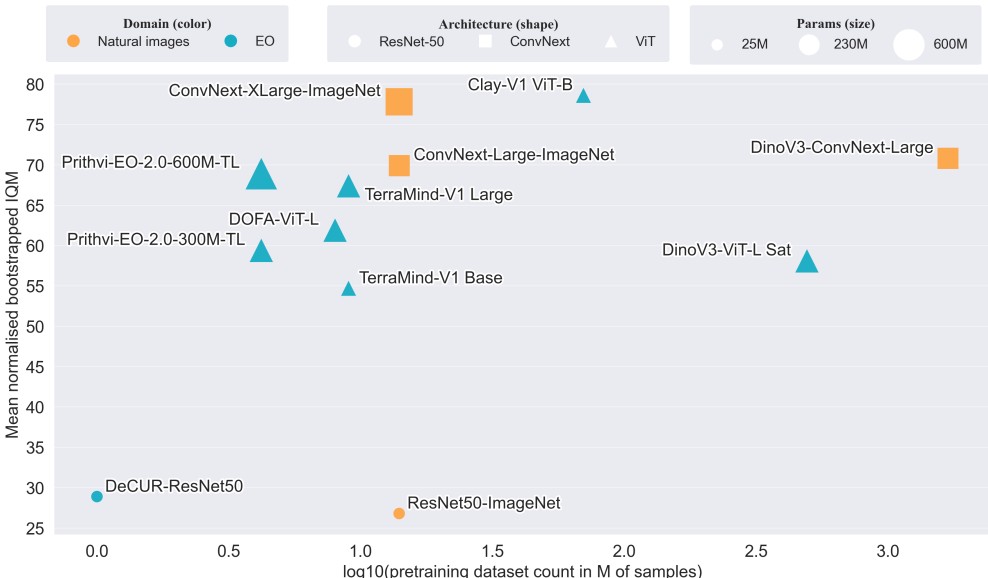

Figure 1: Effect of pretraining factors on model performance on the Core capability.

**Challenging and Discriminative.** Datasets must clearly distinguish strong GeoFMs from baseline models. We experimentally validated discriminative power across multiple models, discarding datasets with overlapping performance profiles in which inter-model differences fell below the variability across random seeds. For example, most well-trained models achieve above 98% accuracy on EuroSAT (Helber et al., 2019) or saturate on multi-modal datasets like Sen1Floods11 (Bonafilia et al., 2020), a pattern previously documented in GEO-Bench (Lacoste et al., 2023) and consistently confirmed in recent GeoFM evaluations (Szwarcman et al., 2024; Jakubik et al., 2025; ClayFoundationModel).

**Open Licenses.** We prioritized permissive licenses to enable academic and industry adoption, avoiding GPL and non-commercial licenses.[1]

**Diversity.** The benchmark encompasses a wide range of tasks, modalities, and geographic regions, featuring samples from all seven continents (Figure 2). The comparatively higher representation of Europe stems from initiatives such as INSPIRE and Horizon Europe (see Figure 24 for relative continental coverage).[2]

Where needed, datasets were sub-sampled to reduce computational cost and formatted following the TACO convention (Aybar et al., 2025), ensuring FAIR compliance (Wilkinson et al., 2016) with self-contained, ML-ready samples. Sub-sampling caps were 20,000 samples for classification and 5,000 for pixel-wise and detection tasks, with the exception of BioMassters.

### 3.2 Capability Groups

Aggregating performance across heterogeneous datasets into a single overall ranking obscures important model characteristics. A GeoFM may excel at high-resolution RGB classification yet underperform on multi-temporal crop mapping or multi-spectral segmentation. Reporting only aggregate metrics therefore limits scientific insight and practical model selection. To address this limitation, GEO-Bench-2 organizes datasets into nine overlapping **capability groups** (Table 3), each isolating a distinct challenge in geospatial modeling. The groups are intentionally overlapping since a dataset may contribute to multiple capabilities if it tests several dimensions simultaneously. We categorize the capability groups as follows:

---

[1]For some capabilities (e.g., detection), finding suitable open-license datasets proved challenging.
[2]To enhance global balance, we encourage the community to release openly licensed dataset datasets from underrepresented regions.

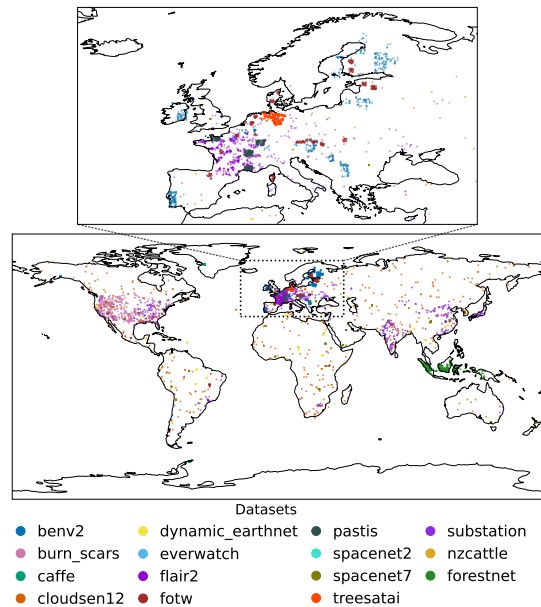

Figure 2: Global distribution of samples (500 random locations per dataset shown for visualization). The BioMassters and So2Sat datasets were originally released without geospatial information and are hence not represented.

| Task | Dataset | Domain | Modalities | GSD | Train/Val/Test | Classes | License |
|---|---|---|---|---|---|---|---|
| Classification | BEN V2 (Clasen et al., 2024) | Land cover | S1+S2 | 10m | 20000/4000/4000 | 19 | CDLA-Permissive-1.0 |
| | TreeSatAI (Ahlswede et al., 2023) | Tree Species | S2 TS | 10m | 20000/4000/4000 | 13 | CC-BY-4.0 |
| | So2sat (Zhu et al., 2020; Lacoste et al., 2023) | Climate Zones | S2 | 10m | 19992/986/986 | 17 | CC-BY-4.0 |
| | Forestnet (Irvin et al., 2020; Lacoste et al., 2023) | Tree Species | L8 | 15m | 6464/989/989 | 12 | CC-BY-4.0 |
| Pixel Regression | BioMassters (Nascetti et al., 2023) | Biomass estim. | S1+S2 TS | 10m | 4011/1739/2776 | $\mathbb{R}_+$ | CC-BY-4.0 |
| Semantic Segmentation | CaFFe (Gourmelon et al., 2022) | Glacier zones | S1 SAR | 10m | 4000/1000/2000 | 4 | CC-BY-4.0 |
| | CloudSen12 (Aybar et al., 2024) | Cloud/shadow | S1+S2 | 10m | 4000/535/975 | 4 | CC0 |
| | NASA Burn Scars (Szwarcman et al., 2024) | Burn scars | HLS | 30m | 524/160/120 | 2 | CC-BY-4.0 |
| | Dynamic EarthNet (Toker et al., 2022) | LULC (temporal) | Planet | 3m | 700/100/200 | 7 | CC-BY-4.0 |
| | FLAIR 2 (Garioud et al., 2023b) | LULC | Aerial RGBN+DEM | 0.2m | 4049/1022/3022 | 13 | Open License 2.0 |
| | FTW (Kerner et al., 2025) | Field boundaries | S2 | 10m | 4000/1000/2000 | 2 | CC-BY-SA |
| | KuroSiwo (Bountos et al., 2024) | Flood extent | S1+DEM+Slope | 10m | 4000/1000/2000 | 4 | MIT |
| | PASTIS (R) (Sainte Fare Garnot et al., 2022) | Crop type map. | S1+S2 | 10m | 1455/482/496 | 19 | CC-BY-4.0 |
| | SpaceNet2 (Van Etten et al., 2018) | Building | Worldview | 0.3m | 5186/1461/2961 | 2 | CC-BY-SA-4.0 |
| | SpaceNet7 (Van Etten et al., 2018) | Building | Planet | 3m | 3888/652/1152 | 2 | CC-BY-SA-4.0 |
| Object Detection | EverWatch (Garner et al., 2024) | Bird species | Aerial RGB | 0.1m | 4429/500/196 | 9 | CC0 |
| | m-nzcattle (Abuaiadah & Switzer, 2022) | Cattle | Aerial RGB | 0.1m | 524/66/65 | 2 | CC-BY-4.0 |
| Instance Segmentation | PASTIS (R) panoptic (Sainte Fare Garnot et al., 2022) | Crop type map. | S1+S2 | 10m | 1455/482/496 | 19 | CC-BY-4.0 |
| | Substations (Jindgar & Lindsay, 2024) | Substations | S2 | 10m | 4000/500/500 | 2 | CC-BY-4.0 |

Table 2: GEO-Bench-2 dataset overview grouped by task category. S1: Sentinel-1; S2: Sentinel-2; DEM: Digital Elevation Model; TS: Time series; LULC: Land Use Land Cover.

**Task-Centric Capabilities.** These capabilities isolate architectural and decoding challenges associated with pretrained model representations:

(1) **Classification:** single- and multi-label image-level prediction.
(2) **Pixel-wise:** semantic segmentation and regression with per-pixel outputs.
(3) **Detection:** object detection and instance segmentation using bounding boxes and masks. Because detection pipelines are computationally heavier and methodologically distinct, these datasets appear in fewer subsets to preserve accessibility of other capabilities.

Separating task types exposes how encoder architectures interact with decoder design and inductive biases.

**Input-Centric Capabilities.** These capabilities evaluate robustness to variations in input structure, modality, and acquisition characteristics:

(4) **Multi-temporal:** datasets with multiple timestamps, designed to measure a model's ability to exploit temporal structure. Models with dedicated temporal architectures are expected to leverage them on this capability.

(5) **<10m Resolution:** high-resolution imagery below 10 m GSD.

(6) **≥10m Resolution:** medium-resolution imagery at or above 10 m GSD.

(7) **RGB/NIR:** datasets limited to visible and near-infrared bands.

(8) **Multi-spectral-dependent:** datasets where additional spectral bands are necessary for competitive performance, validated via ablation (Appendix E.3).

Five datasets (KuroSiwo, PASTIS-R, CaFFe, BEN V2, CloudSEN12) additionally permit isolated evaluation of SAR inputs, enabling modality-specific analysis beyond optical imagery.

**Core Capability.** The final capability provides a computationally efficient yet discriminative subset spanning all major challenges:

(9) **Core:** a balanced subset spanning all task types (classification, pixel-wise, detection), both resolution ranges, and both spectral profiles. Datasets were selected by applying the same discriminative-power criterion from Section 3.1, with the additional constraint that no single task or capability is over-represented, ensuring aggregate Core scores reflect broad model competence rather than strength in any one dimension.

Together, these nine capabilities transform evaluation from a single ranking into a multi-dimensional performance profile. Rather than asking which model is "best" overall, GEO-Bench-2 asks: *best for which capability?* This shift enables targeted model development and more informed deployment decisions.

| Dataset | Classi-fication | Pixel wise | Detection | Multi Temporal | < 10m Res | ≥10m Res | RGB/ NIR | Multi Spectral | Core |
|---|---|---|---|---|---|---|---|---|---|
| BEN V2 | ✓ | | | | | ✓ | | ✓ | ✓ |
| TreeSatAI | ✓ | | | | ✓ | | ✓ | | ✓ |
| So2Sat | ✓ | | | | | ✓ | | ✓ | |
| ForestNet | ✓ | | | | | ✓ | | | |
| BioMassters | | ✓ | | ✓ | | ✓ | | ✓ | ✓ |
| CaFFe | | ✓ | | | | ✓ | | | |
| CloudSEN12 | | ✓ | | | | ✓ | | ✓ | ✓ |
| NASA Burn Scars | | ✓ | | | | ✓ | | ✓ | ✓ |
| Dynamic Earth Net | | ✓ | | ✓ | ✓ | | ✓ | | |
| FLAIR 2 | | ✓ | | | ✓ | | ✓ | | ✓ |
| FTW | | ✓ | | | | ✓ | ✓ | ✓ | ✓ |
| KuroSiwo | | ✓ | | ✓ | | ✓ | | | ✓ |
| PASTIS | | ✓ | | ✓ | | ✓ | | ✓ | ✓ |
| SpaceNet 2 | | ✓ | | | ✓ | | | ✓ | |
| SpaceNet 7 | | ✓ | | | ✓ | | ✓ | | ✓ |
| EverWatch | | | ✓ | | | | | | ✓ |
| NZCattle | | | ✓ | | | | | | |
| PASTIS (R) panoptic | | | ✓ | | | | | | |
| Substations | | | ✓ | | | | | | ✓ |

Table 3: Overview of dataset capabilities. Columns are ordered to match the enumeration in Section 3.2.

# 4 Evaluation Protocol

The primary objective of GEO-Bench-2 is to measure downstream task performance of GeoFMs under controlled and reproducible conditions. However, adapting a pretrained backbone to diverse EO datasets remains an active research problem, with substantial variation in fine-tuning strategies, decoder design, and hyperparameter optimization. The benchmark therefore serves a dual purpose: (1) to provide a standardized baseline for fair comparison, and (2) to enable methodological innovation in model adaptation.

To balance these goals, we define a **prescriptive yet flexible** protocol. The core components of this protocol are framework-agnostic: dataset splits, evaluation metrics, and IQM-based aggregation can be adopted by any training pipeline. TerraTorch-specific tooling (HPO via TerraTorch Iterate, provided decoder implementations) is offered for convenience and is not a mandatory dependency. Users are free to explore alternative adaptation strategies, but must adhere to clearly defined constraints to ensure comparability. As a general rule, adaptation procedures must be generic and not over-engineered for individual datasets. Users must document their full adaptation protocol using the provided template. If evaluating a new GeoFM, we additionally recommend submitting at least one entry that follows the baseline protocol to facilitate direct comparison.

**Why Fine-Tuning?** Although frozen backbones are supported (with a separate dedicated leaderboard), our experiments (Appendix E.1) show substantial performance degradation and significant ranking shifts under fully frozen settings. Because frozen evaluation underestimates representational capacity for most GeoFMs, we recommend full fine-tuning with hyperparameter selection as the primary evaluation setting.

## 4.1 Hyperparameter Optimization

Hyperparameter selection significantly influences downstream performance. Unconstrained tuning, however, risks dataset-specific overfitting and undermines leaderboard fairness. We therefore regulate both the search space and the search budget.

**Search Space.** Search spaces should be largely uniform across the benchmark. Different task types (e.g., classification vs. segmentation) may use separate search spaces, but each search space must be documented and justified. To discourage per-dataset tuning, a search space must apply to at least four datasets. Leaderboard submissions may be flagged as *over-engineered* if customization is excessive.

**Search Budget.** To balance exploration and computational fairness, the maximum search budget is limited to 16 trials per dataset. The best configuration is selected on the validation set. Bayesian optimization methods are permitted.

**Repeated Evaluation.** To account for stochasticity, the selected configuration must be fine-tuned five times with random seeded experiments and evaluated on the test split using the required metric (Section 4.6).

Our baseline protocol uses Optuna (Akiba et al., 2019) within TerraTorch Iterate to automate hyperparameter search over a fixed space. We tune learning rate and batch size while keeping all other hyperparameters constant. Each trial and repeated run is trained for 50 epochs.

## 4.2 Pre-Processing and Data Augmentation

Pre-processing and augmentation choices can introduce hidden performance gains if tailored to individual datasets. To preserve comparability, pre-processing and augmentation must not use information from the test set and must be uniform across all datasets.

**Baseline Protocol.** We apply per-band Z-score normalization to each model, with statistics estimated exclusively on the training split. For regression datasets, Z-score normalization is also applied to target values. During training, we use standard geometric augmentations including horizontal and vertical flipping. For pixel-wise tasks, images larger than $224 \times 224$ pixels are randomly cropped during training. Validation and testing use tiled inference when necessary to preserve full spatial resolution. Although this normalization scheme may not perfectly match the pretraining distribution of all models, we find that fine-tuning mitigates potential discrepancies.

## 4.3 Base Model Adaptation

Adapting a pretrained backbone to diverse downstream tasks requires task-appropriate decoder design while avoiding dataset-specific engineering. To ensure fair comparison, adaptation methodologies must be generic

and valid for at least four datasets. This constraint encourages transferable architectural decisions rather than per-dataset optimization. Our baseline protocol provides a standardized adaptation strategy across task categories.

**Classification.** For single- and multi-label classification, we attach a single linear layer with softmax activation to the encoder output. This minimal head isolates the representational quality of the backbone while avoiding classification task-specific architectural tuning.

**Pixel-wise Tasks (Segmentation and Regression).** For semantic segmentation and pixel-wise regression, we employ a UNet decoder (Ronneberger et al., 2015) fed with equally spaced encoder features (Marti Escofet et al., 2025). Transformer-based backbones do not natively produce hierarchical feature maps. We therefore apply *LearnedFeatureInterpolation* (available in TerraTorch) to structure encoder outputs before passing them to the decoder. Prior work shows that this approach outperforms alternatives such as UPerNet for ViT-based GeoFMs (Marti Escofet et al., 2025).

**Object Detection and Instance Segmentation.** For detection and instance segmentation, we use Faster R-CNN (Ren et al., 2015) and Mask R-CNN (He et al., 2017), respectively, each equipped with a Feature Pyramid Network (FPN) (Lin et al., 2017). This standard pipeline reflects widely adopted detection practices while maintaining comparability across model families.

This baseline protocol prioritizes architectural consistency over dataset-specific customization. Users may propose alternative adaptation strategies, provided they remain generic and satisfy the benchmark's reproducibility constraints.

## 4.4 Multi-Spectral Bands, SAR, and Multi-Modal Datasets

Geospatial datasets frequently contain information beyond standard RGB imagery, including additional optical bands, Synthetic Aperture Radar (SAR), and multi-modal sensor combinations. Handling these modalities consistently is essential for fair comparison, as modality-specific engineering can impact performance. We therefore allow flexibility in modality handling, while requiring that approaches remain generic across datasets. Our baseline protocol standardizes how additional modalities are incorporated into pretrained backbones.

**Multi-Spectral Bands.** When datasets contain multiple optical bands, all available bands are used provided they are compatible with the model architecture. If an exact spectral match between pretraining and downstream bands is not available, bands are aligned to the closest wavelength (e.g., matching Sentinel-2 bands to WorldView channels). This approach prioritizes maximal information usage while preserving architectural consistency.

**Synthetic Aperture Radar (SAR).** When a model does not natively support SAR inputs, VV and VH polarization bands are mapped to the model's RGB channels in the order VV, VH, VV. This simple replication strategy avoids introducing modality-specific encoder modifications while still enabling evaluation on SAR-inclusive datasets.

**Multi-Modal Datasets.** Several datasets provide both Sentinel-1 (SAR) and Sentinel-2 (optical) imagery. The benefit of joint modality usage remains unclear in prior work (Marsocci et al., 2024). We therefore conducted an ablation (Appendix E.5) on three multi-modal models (TerraMind, DOFA, Clay), comparing S2-only performance to S1+S2. For the main benchmark results, we report the best-performing modality configuration per model-dataset pair. In practice, this resulted in using both modalities only for TerraMind on BEN V2 and BioMassters.

This protocol ensures consistent treatment of additional modalities while allowing models designed for multi-modal pretraining to leverage their intended strengths.

### 4.5 Multi-Temporality

For models without dedicated temporal processing, each timestamp is passed through the encoder separately, and the resulting representations are averaged before the decoder. This defines the reference protocol for non-temporal models. Models with dedicated temporal heads (e.g., architectures with learned temporal attention or recurrent modules) are encouraged to use their native temporal processing instead, and should document this in their leaderboard submission. This distinction is intentional: the Multi-Temporal capability is designed to reward models that natively understand temporal structure, while the reference protocol provides a comparable baseline for models that do not.

### 4.6 Evaluation Metrics

All submissions must report results using the following standardized metrics:

- Semantic Segmentation: Multiclass Jaccard Index
- Single-label Classification: Accuracy
- Multi-label Classification: F1-score
- Pixel-wise Regression: Root Mean Squared Error (RMSE)
- Instance Segmentation and Object Detection: Mean Average Precision (mAP)

### 4.7 Aggregated Performance

To obtain a robust aggregation of model performance while accounting for the uncertainty introduced by random seeds, we follow the methodology introduced in Lacoste et al. (2023). We recommend using the leaderboard or the provided code to compute aggregated scores from raw results.

**Renormalization.** Aggregating scores across multiple datasets using simple averages can distort comparisons, where strong performance on one dataset may overshadow smaller improvements elsewhere (Demšar, 2006). To mitigate this, we normalize each dataset's scores to the range $[0, 1]$ via a linear transformation anchored to the worst and best results from a fixed reference model set. The reference set comprises all models evaluated on every dataset in our main experiments (Table 4), chosen to span architectural families (ResNet, ConvNeXt, Swin, ViT), parameter scales (25M to 600M), pretraining domains (natural-image and EO-specific), and the observed performance range. This ensures the $[0, 1]$ bounds reflect a representative envelope of model behavior on each dataset rather than the idiosyncrasies of any single model family. This approach is chosen over rank-based aggregation, which can disrupt leaderboard continuity as ranks shift when new models are added. The $[0, 1]$ range also enhances interpretability compared to z-score normalization. Normalization factors are published in our Anonymous-Bench repository and must be reused for consistency. For completeness, a rank-based table can be found in Appendix A. In addition, we have also included performance profiles (Dolan & Moré, 2002) in Appendix B which do not rely on a single reference model.

**Bootstrap.** We perform a stratified bootstrap over the 5 normalized repeats for each dataset, using 100 resamples.

**IQM Aggregation.** For each normalized bootstrap iteration, results across datasets within a capability are aggregated using the interquartile mean (IQM). This reduces the influence of outliers and provides a more statistically stable estimate (Agarwal et al., 2021).

**Final Aggregation.** The final score and standard deviation are obtained by averaging the results across all 100 normalized IQM bootstrap iterations. Bootstrapped confidence intervals quantify uncertainty due to score variability across runs and datasets. Models whose intervals overlap should not be interpreted as definitively ranked; the benchmark is designed to surface capability-level strengths rather than impose a strict global ordering. Pairwise significance tests are not applied at the capability level because aggregating across heterogeneous datasets, tasks, and metrics does not yield a well-defined pairing unit, making bootstrapped

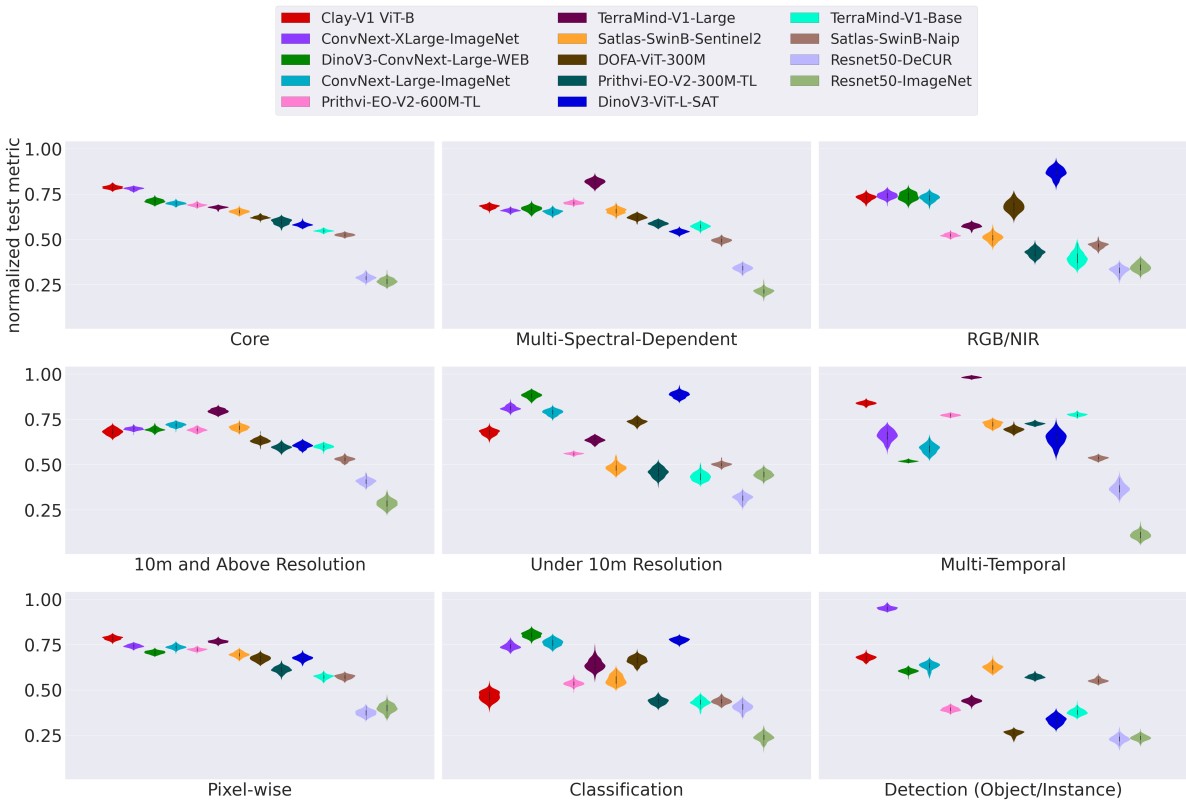

Figure 3: Normalized bootstrapped IQM performance by capability. Models are ordered by Core capability score. Overlapping confidence intervals indicate that the relative ordering of closely ranked models is uncertain; such models should not be treated as definitively ranked.

CIs the appropriate uncertainty measure. This differs from the ablation studies in Appendix E, where paired tests are appropriate because the same model is evaluated under two controlled conditions on identical data.

### 4.8   Leaderboard

To foster a community effort, we provide a leaderboard and encourage users to submit experiments (both successful and negative results) and to reproduce existing experiments. Users may submit an entry for a single capability without evaluating all datasets. For the Multi-Temporal capability, submissions must indicate whether the model uses a dedicated temporal processing mechanism or the reference average-pooling protocol, to enable fair interpretation of results. We reserve the right to reject submissions that: do not follow the required guidelines, cannot be reproduced (e.g., closed-source model), exhibit per-dataset over-engineering, or undermine the main objectives of the benchmark.

## 5   Results

We report normalized bootstrapped IQM scores across all capability groups in Figure 3. Models in 1 are ordered by Core capability performance. Raw per-dataset metrics are provided in Appendix D.1. The results reveal that GeoFM evaluation is inherently multi-dimensional: model rankings vary substantially across capabilities, confirming that aggregate scores alone are insufficient to characterize performance.

| Model | Type | # Backbone Params | Learning Technique | Data | Res | N | T | License |
|---|---|---|---|---|---|---|---|---|
| Resnet50-ImageNet | ResNet-50 | 25M | Supervised | ImageNet-22k | NA | 14M | 1 | Apache 2.0 |
| ConvNext-Large-ImageNet (Wightman, 2019) | ConvNext | 230M | Supervised | ImageNet-22k | NA | 14M | 1 | Apache 2.0 |
| ConvNext-XLarge-ImageNet (Wightman, 2019) | ConvNext | 390M | Supervised | ImageNet-22k | NA | 14M | 1 | Apache 2.0 |
| DINOv3-ViT-L-SAT (Siméoni et al., 2025) | ViT | 300M | Distillation | Maxar RGB | 0.6 m | 493M | 1 | DINO V3 |
| DINOv3-ConvNext-Large-WEB (Siméoni et al., 2025) | ConvNext | 230M | Distillation | LVD-1689M | NA | 1689M | 1 | DINO V3 |
| Resnet50-DeCUR (Wang et al., 2024) | ResNet-50 | 25M | Contrastive | Sentinel-2 | 10 m | 1M | 1 | Apache 2.0 |
| DOFA-ViT-300M (Xiong et al., 2024) | ViT | 300M | MAE | S1, S2, EnMap, Gaofen, Landsat | 1–30 m | 8M | 1 | CC-BY-4.0 |
| Clay-V1 ViT-B (ClayFoundationModel) | ViT | 86M | MAE | LS8&9, S1&2, NAIP, LINZ, MODIS | 1–30 m | 70M | 1 | Apache 2.0 |
| Satlas-SwinB-Sentinel2 (Bastani et al., 2023) | Swin | 88M | Supervised | Sentinel-2 | 10 m | NA | 1 | ODC-BY |
| Satlas-NAIP (Bastani et al., 2023) | Swin | 88M | Supervised | NAIP | 1 m | NA | 1 | ODC-BY |
| Prithvi-EO-V2-300M-TL (Szwarcman et al., 2024) | ViT | 300M | MAE | HLS | 30 m | 4.2M | 4 | Apache 2.0 |
| Prithvi-EO-V2-600M-TL (Szwarcman et al., 2024) | ViT | 600M | MAE | HLS | 30 m | 4.2M | 4 | Apache 2.0 |
| TerraMind-V1-Base (Jakubik et al., 2025) | ViT | 86M | Correlation | S1&2, LULC, DEM, NDVI | 10 m | 9M | 1 | Apache 2.0 |
| TerraMind-V1-Large (Jakubik et al., 2025) | ViT | 300M | Correlation | S1&2, LULC, DEM, NDVI | 10 m | 9M | 1 | Apache 2.0 |

The pretraining dataset size ($N$) is estimated where not clearly reported. Checkpoint sources: Torchgeo (DOFA, DeCUR, Satlas); TerraTorch (Prithvi, TerraMind); Timm (ResNet50-ImageNet, ConvNext-ImageNet); Meta (DINO V3).

Table 4: Characteristics of the benchmarked models: backbone type (Type), number of backbone parameters (# Param.), pretraining technique (Technique), pretraining data (Data), spatial resolution (Res.), number of samples ($N$), and timestamps per sample ($T$). Satlas characteristics refer to the Sentinel-2 pretrained checkpoint.

## 5.1 Models Used in Our Experiments

To probe the benchmark, we selected models spanning diverse architectures, parameter scales, and pretraining paradigms (Table 4). These include: (i) supervised ImageNet models, (ii) self-supervised and distillation-based natural image models and (iii) EO-specific foundation models pretrained on multi-spectral or multi-temporal satellite data. Although this selection of models covers a wide array of the most commonly used models in the remote sensing domain, we emphasize that it is not exhaustive and that conclusions drawn from our results below should be contextualized in that regard. All main experiments use full end-to-end fine-tuning, with pretrained backbones unfrozen during adaptation, following our recommended evaluation protocol (Section 4).

## 5.2 General Trends Across Capabilities

No single model dominates across all nine capabilities. Instead, rankings shift depending on task formulation and input characteristics.

EO-specialized models such as TerraMind and Prithvi consistently lead multi-spectral and multi-temporal capabilities, while natural-image pretrained models such as DINOv3-ViT-L-SAT and ConvNeXt variants excel in high-resolution RGB/NIR settings. This divergence validates the motivation behind capability-based evaluation: models encode biases from pretraining data, pretraining task, architecture choices and model sizes that translate differently across downstream regimes.

### 5.3 Impact of Architecture, Size, and Pretraining Dataset

Figure 1 relates Core capability performance to architecture type, parameter count, and pretraining dataset size. Model scale generally correlates with improved performance, particularly within architecture families. However, scale alone does not explain outcomes. Clay-V1 ViT-B achieves top Core performance with only 86M parameters, likely due to its diverse EO-specific pretraining corpus ($\sim$70M heterogeneous samples spanning 1–30 m GSD) and smaller patch size (8), which preserves fine-grained spatial detail. In contrast, ResNet-50 variants perform poorly across capabilities, indicating that architectural capacity and representational flexibility remain critical. Notably, ConvNeXt models pretrained on natural images adapt effectively to EO tasks under full fine-tuning. This suggests that high-capacity architectures pretrained at scale can compensate for domain mismatch when adaptation is sufficiently expressive.

### 5.4 Importance of Multi-Spectral Bands

The RGB/NIR and < 10m GSD capabilities are dominated by DINOv3 and ConvNeXt models, reflecting their strong performance on high-resolution optical imagery. In contrast, EO-specific models take the lead in the Multi-Spectral-Dependent capability. Although aggregate normalized differences appear moderate, dataset-level analysis (Appendix D.1) reveals substantial gaps. On NASA Burn Scars and PASTIS crop classification, EO-specialized models outperform RGB-only models by up to 10%, demonstrating the practical importance of additional spectral bands. These findings confirm that multi-spectral information remains essential for certain EO tasks, particularly those involving vegetation, land cover discrimination, or environmental monitoring. The capability grouping therefore captures a meaningful and operationally relevant distinction.

### 5.5 Model Performance Across Task Types

Within the evaluated model families and tasks, inductive biases appear to interact with task formulation. Clay-V1-ViT-B leads the Pixel-wise capability, consistent with its smaller patch size and improved spatial detail retention. Conversely, DINOv3 and ConvNeXt variants perform best in Classification and Detection. For detection tasks, convolutional inductive biases appear advantageous: most ViT-based GeoFMs underperform relative to ConvNeXt-based models when paired with Faster R-CNN or Mask R-CNN. This suggests that standard detection heads may better exploit hierarchical convolutional features than token-based representations. Overall, these patterns highlight that architectural suitability is task-dependent. Capability-based evaluation makes these trade-offs explicit, enabling informed selection rather than reliance on a single aggregate ranking.

### 5.6 Ablations

Beyond the primary benchmark configuration, we conducted controlled ablations to assess the sensitivity of model performance and rankings to key methodological choices. Ranking changes are quantified using the normalized Kendall tau distance, defined as the fraction of model pairs whose order changes under an alternative setting (Figure 4). We highlight three ablation findings, while referring the reader for additional ablation studies to Appendix E.

**Frozen vs. Fine-Tuned Backbones.** Contrary to findings of Marsocci et al. (2024), freezing the encoder produces a systematic drop in performance across nearly all datasets (Appendix E.1). More importantly, freezing induces ranking changes in over 20% of model pairs. Our findings suggest that this methodological choice can lead to different model selection recommendations for practitioners depending on their available computational budget. However, large gaps between frozen and unfrozen combinations also demonstrate that significant room for improvement remains for GeoFM representation expressiveness.

**Decoder Complexity.** Replacing the UNet decoder with a linear head for pixel-wise tasks consistently reduces performance (Appendix E.2), particularly on datasets requiring fine-grained boundary delineation such as FTW and SpaceNet7. Although ranking shifts are smaller than in the frozen ablation, this experi-

ment highlights that decoder choice materially affects evaluation outcomes and that the inductive biases of convolution-based decoders are useful. Standardizing decoder design is therefore critical for fair benchmarks.

**Multi-Temporal Processing.** Using a single timestamp instead of multiple temporal views leads to marked performance drops on datasets such as PASTIS (Appendix E.6). Among all ablations, this setting produces the largest ranking perturbations aside from freezing. This finding underscores that temporal modeling remains an underexploited but decisive factor in GeoFM performance. It also confirms the necessity of explicitly isolating multi-temporal capability in the benchmark.

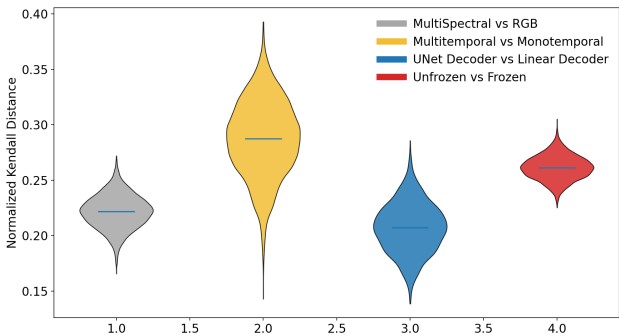

Figure 4: Normalized Kendall tau distance showing the fraction of model pairs with altered ranking between two methods. Uncertainty was computed via 2,000 bootstrap resamples.

# 6 Discussion and Limitations

Our results validate the central premise of GEO-Bench-2: GeoFM performance is inherently multi-dimensional and cannot be faithfully summarized by a single aggregate score. Foundation models for EO are advancing rapidly (Zhu et al., 2024), with strong appeal for domain-specific applications. However, the field has yet to experience a paradigm shift comparable to that of generative modeling in text or computer vision. While our analysis confirms that larger models tend to perform better, we do not observe broad, cross-task scaling behavior or emergent effects of the kind reported for LLMs or image generation (Hoffmann et al., 2022; Li et al., 2024).

The LLM community benefits from established benchmarks and centralized infrastructure for tracking progress (White et al., 2024). With GEO-Bench-2, we provide an analogous framework for the EO domain. A key advantage of the leaderboard is enabling users to rank models across capabilities and settings, allowing the community to identify models suited to specific use cases and resource constraints.

Recent work has shown that models pretrained on natural images remain competitive on EO tasks (Corley et al., 2024), well-tuned supervised models can match specialized FMs (Xu et al., 2025), and most GeoFMs struggle under frozen encoder settings (Marsocci et al., 2024), with TerraMind being the first exception (Jakubik et al., 2025). Our experiments tell a more nuanced story: under full fine-tuning, GeoFMs consistently outperformed simpler models (e.g., ResNet-50) but did not consistently outperform larger natural-image models such as ConvNeXt, particularly for high-resolution RGB-only tasks. However, GeoFMs demonstrated clear advantages when multi-spectral information is critical, a setting explicitly captured by GEO-Bench-2. Our results align with DINOv3 findings (Siméoni et al., 2025), where even the 7B RGB-only model could not surpass the 10× smaller Prithvi-EO-2.0 in crop classification.

Rather than discouraging, these findings reveal opportunities. GEO-Bench-2 pinpoints where current GeoFMs underperform and provides a testbed for developing models that leverage EO's unique characteristics: multi-spectral data, temporal dynamics, and global coverage. This can guide the community toward pretraining strategies optimized for multi-spectral and temporal data, as well as unexplored architectural directions (e.g., ConvNeXt in EO) that translate to real gains in disaster response, environmental monitoring, and agriculture at scale.

### 6.1 Limitations

As with any benchmark, GEO-Bench-2 reflects a set of design choices and trade-offs. While we aimed for broad coverage across tasks, modalities, and geographic regions, the benchmark does not exhaust the space of GeoFM evaluation settings. Although we evaluate a diverse set of leading models, additional architectures and pretraining strategies remain unexplored. The leaderboard is intended to support continuous community-driven evaluation. Dataset coverage, despite efforts toward diversity, remains heavily skewed toward Europe and North America, reflecting broader patterns in openly available EO data. This is a central limitation that likely requires a cohesive effort of the entire EO community, so that GeoFMs can be adequately probed for performance in limited data regions. Moreover, the benchmark focuses on standardized task metrics and does not assess predictive uncertainty, calibration, or deployment constraints such as latency or memory footprint.

GEO-Bench-2 targets the GeoFM paradigm: encoder-style models adapted to downstream tasks via fine-tuning. We therefore do not evaluate multimodal large language models (MLLMs) such as GPT-5, Gemini, Qwen-VL, or EO-specialized variants like GeoChat and RemoteCLIP. This exclusion reflects a paradigm and modality mismatch rather than an oversight: MLLMs are predominantly RGB-only, do not natively consume multi-spectral or SAR inputs (although adoption is feasible as shown by Mallya et al. (2025)), cannot produce dense pixel-level outputs required for segmentation and regression, and are typically evaluated under zero-shot or few-shot prompting rather than the fine-tuning protocol that defines this benchmark. Systematic evaluation of MLLMs on EO tasks is the focus of complementary work such as GEOBench-VLM (Danish et al., 2025), which reports that state-of-the-art VLMs achieve limited accuracy on geospatial tasks. Integrating prompting-based evaluation paradigms into capability-driven GeoFM benchmarking is a promising direction for future work.

Following Rolf et al. (2021), we view this benchmark not as a fixed endpoint but as an evolving evaluation framework that should progressively incorporate more realistic and deployment-aware conditions.

### Broader Impact Statement

GEO-Bench-2 and the models evaluated in this work are intended for Earth Observation applications including disaster response, agriculture, and environmental monitoring. While these are overwhelmingly beneficial uses, practitioners should be aware that geospatial AI systems can reproduce biases present in training data or evaluation benchmarks. The geographic imbalance in our benchmark (Section 6) is an example of such a limitation; users deploying these models in underrepresented regions should validate performance independently.

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

## A  Rankings

| | Core | Pixel-wise | Classification | Detection | Multi Temporal | < 10m | >=10m | RGB/NIR | Multi-Spectral |
|---|---|---|---|---|---|---|---|---|---|
| Clay-V1 ViT-B | **78.6** | **78.4** | 45.9 | 67.8 | 83.7 | 67.4 | 68.4 | 72.8 | 68.0 |
| ConvNext-XLarge ImageNet | 77.8 | 74.2 | 73.8 | **95.1** | 65.4 | 81.4 | 69.7 | 74.3 | 65.8 |
| DinoV3-ConvNext Large-WEB | 70.8 | 70.5 | **80.4** | 60.4 | 51.9 | 88.5 | 69.2 | 73.9 | 66.8 |
| ConvNext-Large ImageNet | 69.9 | 73.6 | 75.9 | 63.1 | 58.8 | 78.7 | 72.3 | 72.7 | 65.4 |
| Prithvi-EO-2.0-600M-TL | 68.9 | 72.2 | 53.4 | 39.3 | 77.1 | 56.1 | 69.1 | 52.3 | 70.2 |
| TerraMind-V1 Large | 67.4 | 76.7 | 63.7 | 43.8 | **98.1** | 63.6 | **79.6** | 57.3 | **81.8** |
| Satlas-SwinB Sentinel2 | 65.6 | 69.5 | 55.4 | 62.9 | 72.0 | 48.3 | 70.4 | 50.9 | 65.4 |
| DOFA-ViT-L | 61.9 | 67 | 66.6 | 25.9 | 69.2 | 73.7 | 62.9 | 68.1 | 62.0 |
| Prithvi-EO-2.0-300M-TL | 59.4 | 61.3 | 43.6 | 57.2 | 72.4 | 45.7 | 59.1 | 42.7 | 58.5 |
| DinoV3-ViT-L SAT | 58.1 | 67.4 | 77.2 | 34.1 | 65.2 | **88.6** | 60.2 | **86.9** | 53.9 |
| TerraMind-V1 Base | 54.7 | 57.2 | 43.0 | 37.7 | 77.4 | 43.3 | 59.9 | 39.5 | 57.1 |
| Satlas-SwinB Naip | 52.6 | 57.4 | 43.7 | 55.2 | 53.4 | 50.1 | 52.9 | 46.3 | 49.5 |
| DeCUR-Resnet50 | 28.9 | 36.9 | 40.4 | 22.7 | 37.1 | 31.2 | 40.4 | 32.7 | 34.0 |
| Resnet50-ImageNet | 26.8 | 39.9 | 23.9 | 23.5 | 11.1 | 43.9 | 28.4 | 34.1 | 21.2 |

Table 5: MinMax scores across all capabilities. These scores were used to draw the final ranking for Table 1. Lighter color indicates a higher positional ranking (i.e., better performance) within a capability. Bold highlights top performer while underlined shows second place.

| | Core | Pixel-wise | Classification | Detection | Multi Temporal | < 10m | >=10m | RGB/NIR | Multi-Spectral |
|---|---|---|---|---|---|---|---|---|---|
| Clay-V1 ViT-B | **4.3** | **4.0** | 10.5 | 5.0 | 3.3 | 4.6 | 6.3 | 3.6 | 5.6 |
| ConvNext-XLarge ImageNet | 4.8 | 4.9 | 4.0 | **1.5** | 6.3 | 3.4 | 5.3 | 3.6 | 5.5 |
| ConvNext-Large ImageNet | 5.7 | 5.5 | 4.5 | 4.5 | 7.5 | 4.8 | 5.5 | 5.0 | 6.4 |
| DinoV3-ConvNext Large-WEB | 5.8 | 6.7 | **2.3** | 5.0 | 11.0 | 3.8 | 6.4 | 4.8 | 5.8 |
| Prithvi-EO-2.0-600M-TL | 6.0 | 6.6 | 8.0 | 8.5 | 8.0 | 7.8 | 6.6 | 7.4 | 5.0 |
| TerraMind-V1 Large | 6.3 | 6.1 | 5.5 | 8.0 | **2.0** | 7.4 | **5.2** | 7.6 | **4.5** |
| Satlas-SwinB Sentinel2 | 7.1 | 7.2 | 8.3 | 4.8 | 6.8 | 8.6 | 6.9 | 7.6 | 7.4 |
| DinoV3-ViT-L SAT | 7.2 | 6.5 | 3.3 | 9.5 | 6.0 | **2.8** | 7.0 | **3.4** | 8.4 |
| DOFA-ViT-L | 7.4 | 6.9 | 6.3 | 11.8 | 5.0 | 6.6 | 6.8 | 7.4 | 7.1 |
| Prithvi-EO-2.0-300M-TL | 8.7 | 9.2 | 10.8 | 5.8 | 7.5 | 11.2 | 8.8 | 10.6 | 8.0 |
| Satlas-SwinB Naip | 8.7 | 9.2 | 10.5 | 6.3 | 10.3 | 9.8 | 9.4 | 9.4 | 9.6 |
| TerraMind-V1 Base | 9.1 | 9.7 | 9.0 | 10.5 | 7.0 | 11.4 | 8.6 | 11.2 | 8.0 |
| Resnet50-ImageNet | 11.8 | 10.9 | 11.3 | 11.8 | 13.8 | 11.0 | 11.0 | 11.6 | 12.3 |
| DeCUR-Resnet50 | 12.4 | 11.5 | 11.0 | 12.3 | 10.8 | 11.8 | 11.2 | 11.8 | 11.5 |

Table 6: Position-based Ranking across all capabilities. For each dataset within a capability, model performance was calculated by bootstrapping with replacement 100 times over the five repeated experiments to account for uncertainty. The average performance across the 100 bootstrapped means was then used to establish the ranking for each dataset. These rankings were averaged across datasets pertaining to a specific capability, yielding the scores in the table. Lighter color indicates a higher positional ranking (i.e., better performance) within a capability. Bold highlights top performer while underlined shows second place.

## B  Performance Profiles

Performance profiles summarize robustness across tasks by plotting, for each model, the fraction of tasks where it is within a factor $\tau$ of the best model on that task (Dolan & Moré, 2002; Nathani et al., 2025). Figure 5 shows the resulting performance profiles. The five seeds per backbone are aggregated using the

interquartile mean (IQM) of the test metric. For the BioMassters dataset, RMSE values are converted to a normalized score before aggregation. For each capability dimension, we keep only backbones that have non-missing IQM scores for all datasets in that dimension, and exclude any backbone with a non-positive IQM score on any dataset in that dimension.

We follow the implementation of Nathani et al. (2025) where $s_{b,d}$ denotes the IQM score of backbone $b$ on dataset $d$. For each dataset $d$, we compute the best score $s^d = \max_b s_{b,d}$ and the performance ratio

$$r_{b,d} = \frac{s^d}{\max(s_{b,d}, \epsilon)},$$

with $\epsilon = 10^{-9}$ to avoid division by zero. The performance profile is then defined as

$$\rho_b(\tau) = \frac{1}{|D|} \sum_{d \in D} \mathbf{1}\{r_{b,d} \leq \tau\}.$$

We report the curve $\rho_b(\tau)$ for each backbone, along with the win rate $\rho_b(1)$ (the fraction of tasks on which backbone $b$ matches the best model) and the area under the profile curve (AUP), defined as $\mathrm{AUP}_b = \int_1^{\tau_{\max}} \rho_b(\tau)\, d\tau$, where $\tau_{\max}$ is the smallest $\tau$ at which $\rho_b(\tau) = 1$ for all backbones $b$.

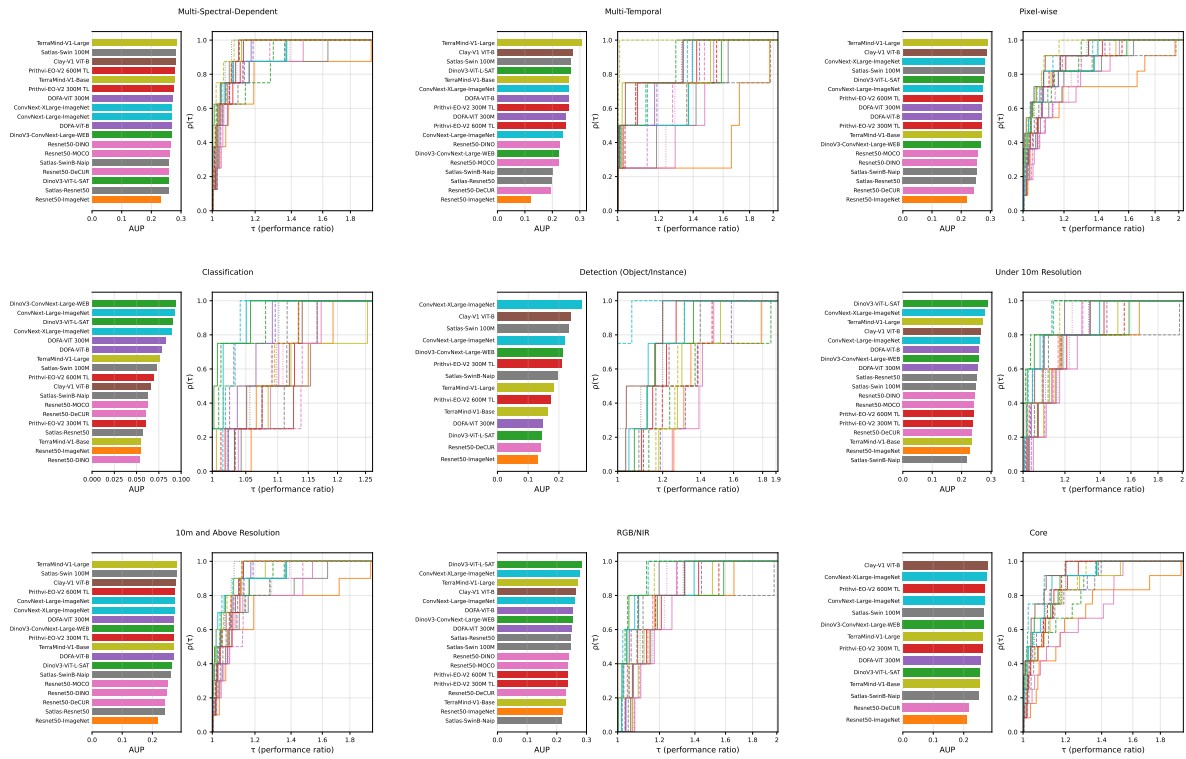

Figure 5: Performance profiles for each capability group. These profiles do not depend on a single reference model set and are insensitive to score scaling. Curves that rise faster and reach higher values indicate models that are consistently near-best across many tasks.

## C   Datasets

The following sections introduce the different datasets as well as the respective preprocessing schemes chosen to generate the benchmark version of the dataset.

### C.1 BigEarthNet V2

BigEarthNet V2 (Clasen et al., 2024) is an updated version of the popular BigEarthNet (Sumbul et al., 2019) multi-label classification dataset with Sentinel 1 and 2 imagery over Europe. The classification labels are derived from the CORINE Land Cover (CLC) labels. In BigEarthNet V2, the Sentinel 2 images were updated through a newer atmospheric correction scheme, the labels were revised with the most recent CLC labels, and a new geographic train, validation, and test split was introduced that reduces their spatial correlation.

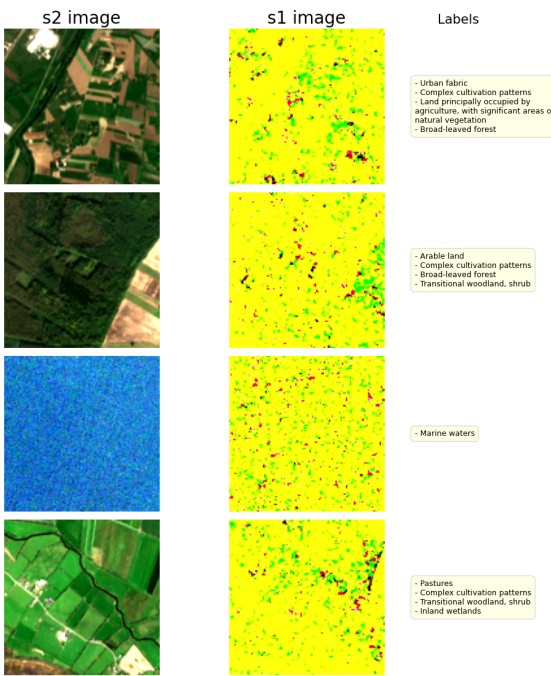

Figure 6: Training Set Examples for BigEarthNet V2 dataset.

### C.2 BioMassters

The BioMassters dataset (Nascetti et al., 2023) consists of multi-temporal Sentinel 1 and 2 imagery over Finland and has pixel-wise above-ground biomass (AGB) annotations that were derived from airborne LiDAR measurements. The dataset was released with a designated train, validation, and test set, but without geospatial information.

### C.3 CaFFe

The Calving Fronts and Where to Find them (CAFFe) dataset (Gourmelon et al., 2022) includes SAR imagery of glaciers from Antarctica, Greenland, and Alaska, with zone segmentation labels for rock, glacier, and ocean or ice melange. In the designated train, validation, and test set, the test set locations are disjoint from the other sets. The dataset was released as single-channel PNG files, and was reprocessed into TIF files with location information merged from a metadata table.

### C.4 CloudSen12

Introduced by Aybar et al. (2024), the CloudSen12+ dataset forms a comprehensive global collection of Sentinel-1 and Sentinel-2 imagery for cloud and shadow detection. Samples were assigned to a train, validation, or test set based on a spatially stratified block split strategy. For our benchmark, we only select the processed samples with 512x512 pixels and the "high-quality" label tag.

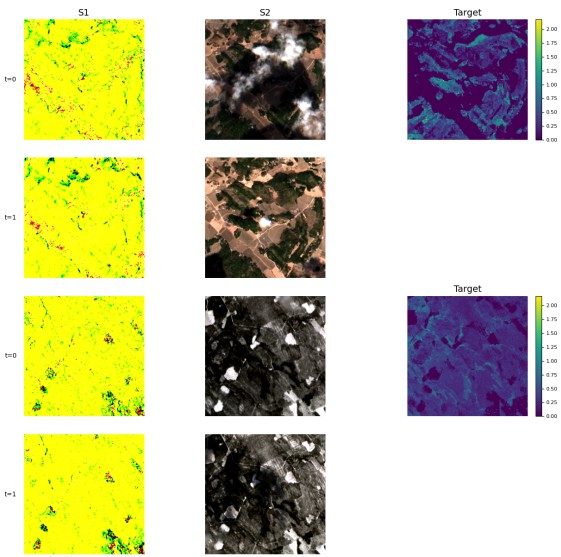

Figure 7: Training Set Examples for Biomassters dataset.

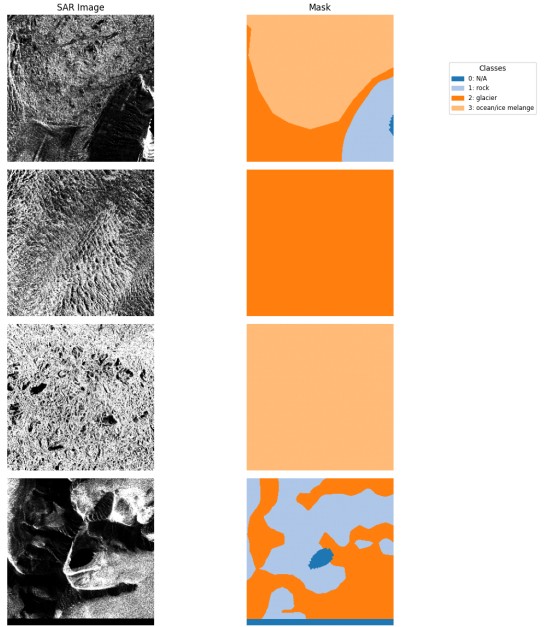

Figure 8: Training Set Examples for CaFFe dataset.

### C.5    DynamicEarthNet

The DynamicEarthNet dataset (Toker et al., 2022) is a global multi-temporal dataset of high-resolution Planet Lab imagery and Sentinel-2 imagery with seven land use and land cover (LULC) classes. Based on 75 different areas of interest, across six continents, the dataset provides daily Planet imagery across these locations from 2018 to 2020. The publicly available subset was processed as follows. First, the 1024x1024 tiles were split into four 512x512 patches. Subsequently, an eight-by-eight grid binning strategy was used to assign samples to a train, validation or test set. The resulting splits do not overlap in either space or time.

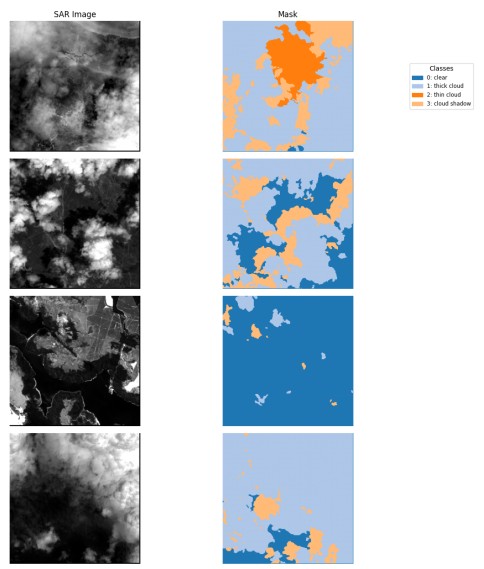

Figure 9: Training Set Examples for CloudSen12 dataset.

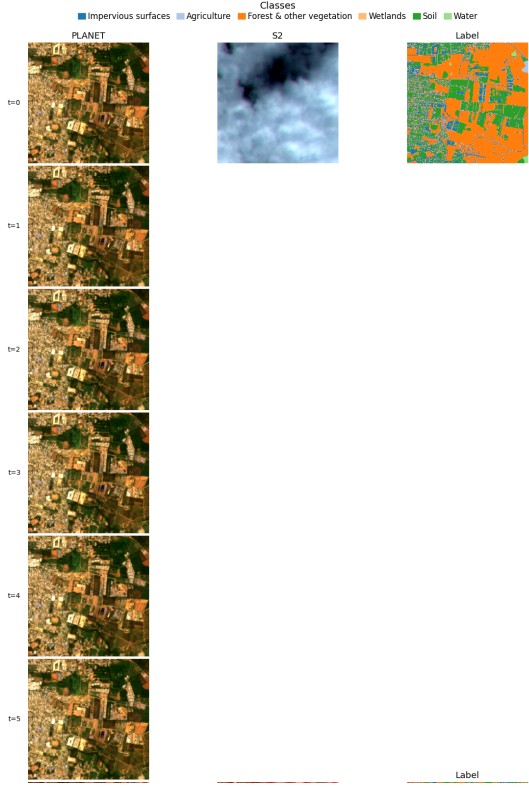

Figure 10: Training Set Examples for DynamicEarthNet dataset.

## C.6 EverWatch

EverWatch (Garner et al., 2024) is a bird detection dataset with high-resolution imagery over the Everglades region. The aerial drone imagery was manually annotated with seven different bird species labels.

The dataset comes with a designated train, validation, and test set and only partially complete geospatial information. The variable-sized tiles were resized to patches of size 512x512.

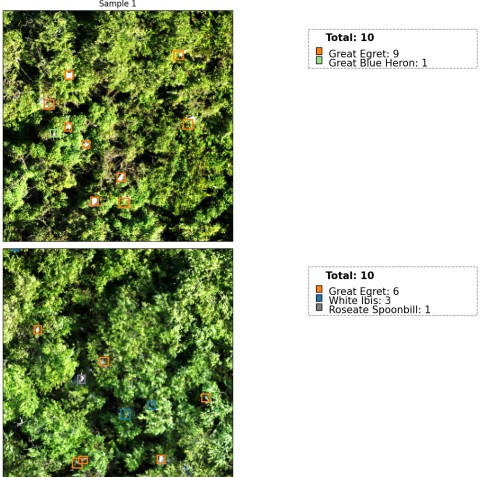

Figure 11: Training Set Examples for EverWatch dataset.

## C.7 Fields of The World

The Fields of The World (FTW) dataset (Kerner et al., 2025) is a global dataset of Sentinel-2 imagery for agricultural field boundary delineation. Each sample consists of two temporal views from different time

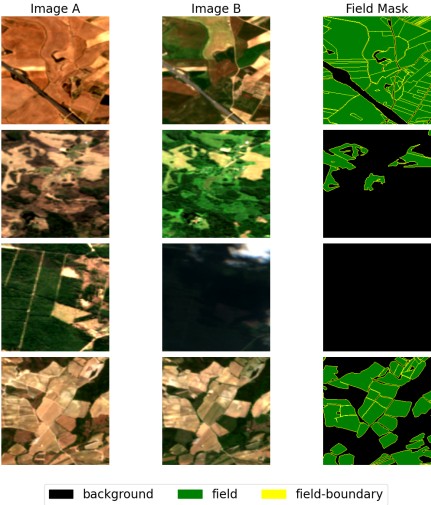

Figure 12: Training Set Examples for FTW dataset.

stamps and a field mask. The dataset was filtered by country to only include data consistent with the open CC-BY-SA license. For this subset, the original train, validation, and test split was used.

## C.8 FLAIR2

The French Land cover from Aerospace ImageRy (FLAIR) Version 2 dataset (Garioud et al., 2023a) is an updated version of FLAIR (Garioud et al., 2022) that now also contains Sentinel-2 time-series data in addition to the high-resolution aerial imagery with thirteen semantic land cover classes across France. The new version also includes an updated test split from distinct spatial domains across the country.

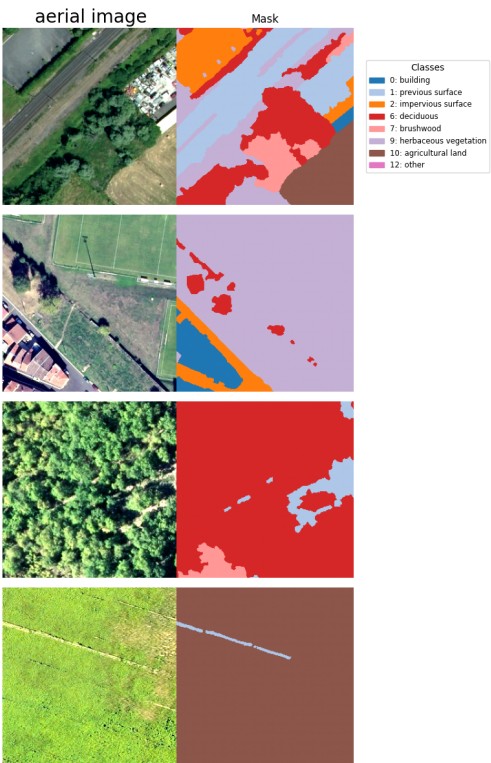

Figure 13: Training Set Examples for FLAIR2 dataset.

## C.9 KuroSiwo

The KuroSiwo dataset (Bountos et al., 2024) is a global dataset of SAR imagery from a broad range of global flooding events with annotations for permanent water and flood water. We use the designated train, validation, and test set.

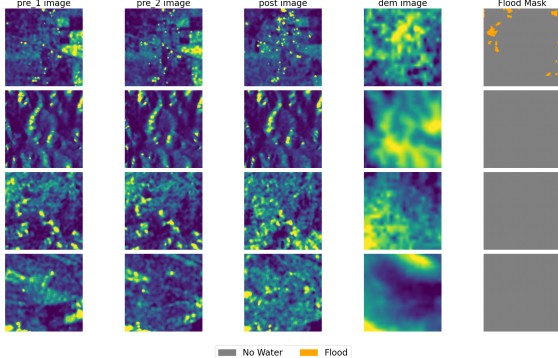

Figure 14: Training Set Examples for KuroSiwo dataset.

## C.10 PASTIS-R

The Panoptic Agricultural Satellite Time Series (PASTIS) dataset (Garnot et al., 2022) is a multi-temporal dataset with Sentinel-1 SAR and Sentinel-2 imagery with crop type annotations across France. We utilize the PASTIS-R version, which is a superset of the original PASTIS dataset and includes the SAR imagery. The dataset comes with a designated train, validation, and test split.

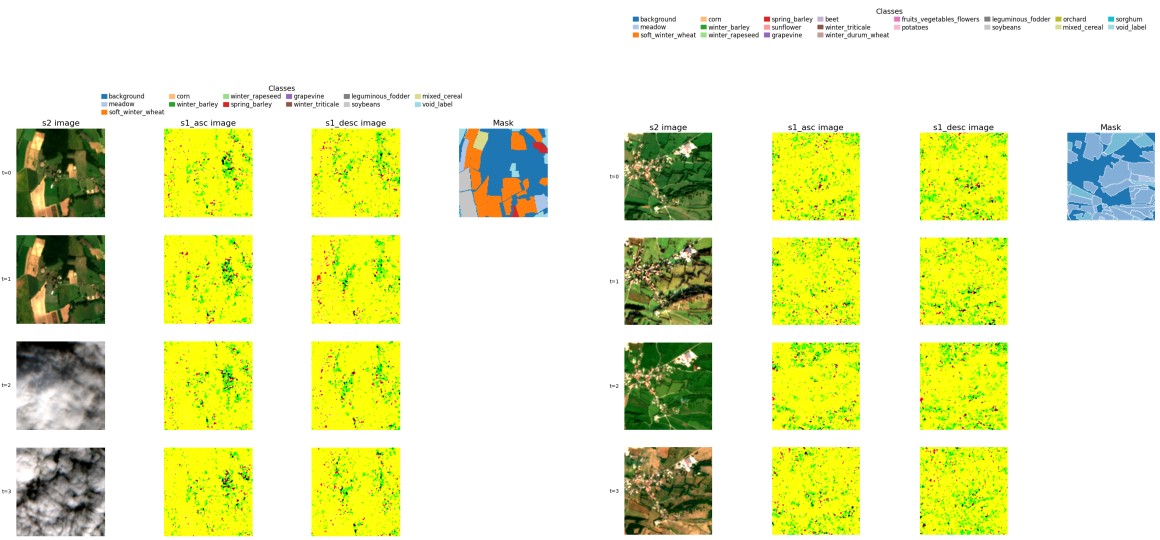

(a) Training Set Examples for PASTIS Semantic Segmentation dataset.

(b) Training Set Examples for PASTIS Instance Segmentation dataset.

Figure 15: Training Set Examples for PASTIS datasets.

### C.11 SpaceNet2

The second edition of the SpaceNet dataset series (Van Etten & Hogan, 2021) consists of high-resolution aerial imagery across four different cities (Las Vegas, Paris, Shanghai, Khartoum) with binary building footprint annotations. As a competition dataset, the publicly available data that includes annotations is only designated for training. To have separate training and evaluation sets, we use the following procedure: For each city, we use a checkerboard style split, inspired by MOSAIKS (Rolf et al., 2021), that overlays a grid structure and all samples within a grid are assigned to a train, validation and test set such that their percentage of the total samples is roughly 70-20-10 respectively.

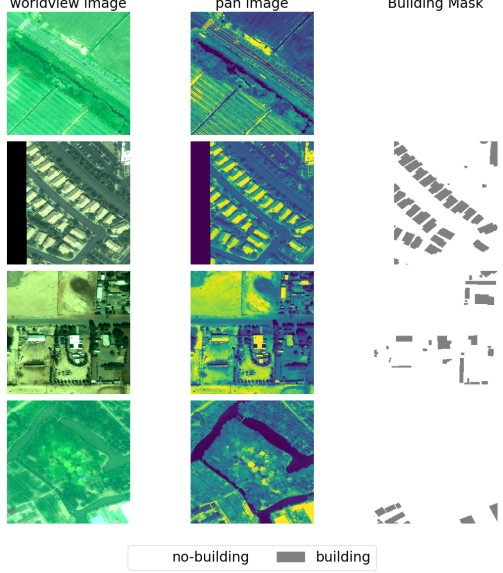

Figure 16: Training Set Examples for SpaceNet2 dataset.

### C.12 SpaceNet7

The seventh edition of the SpaceNet dataset series (Van Etten et al., 2021) is a multi-temporal building footprint dataset of 101 different locations across the globe. For each location, there is a sequence of Planet Lab Dove RGB imagery with fine-grained building annotations. While the intended purpose was to track building changes over time, we cast it as a static building segmentation dataset. The original 1024x1024 tiles were separated into 4 512x512 patches, and samples were assigned to a train, validation, or test split such that their locations are disjoint and their percentage of total samples is roughly 70-20-10 respectively.

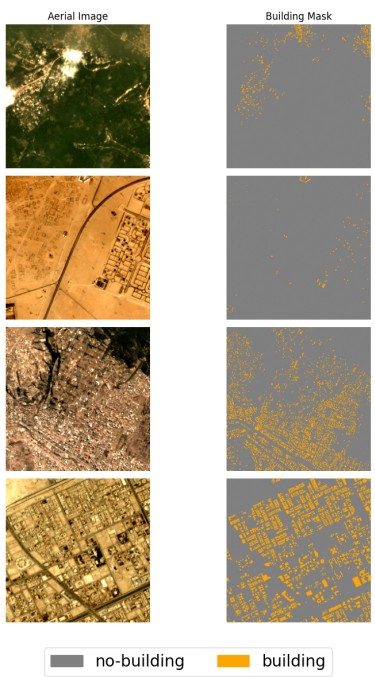

Figure 17: Training Set Examples for SpaceNet7 dataset.

### C.13 TreeSatAI

The TreeSatAI dataset (Ahlswede et al., 2023) originally included single-timestamp aerial, Sentinel-1, and Sentinel-2 imagery with corresponding multi-label tree species classes. Astruc et al. (2024) extended the dataset to include a time series for the Sentinel modalities.

The samples were assigned to a train, validation, or test set with a checkerboard style split strategy, where a 10x10 grid was overlaid on the dataset area of northern Germany, and samples within each block belong to one split such that there is roughly a 70-20-10 distribution across splits.

### C.14 NZCattle

The NZCattle is a dataset included in GEO-Bench (Lacoste et al., 2023). It includes high-resolution aerial RGB images to detect cattle in New Zealand. Originally, the dataset was presented as a segmentation task, which we repurpose here as an object detection task. Original splits were retained.

### C.15 BurnScars

This dataset contains Harmonized Landsat and Sentinel-2 imagery of burn scars and the associated masks for the years 2018-2021 over the contiguous United States (Szwarcman et al., 2024). There are 804 512x512 images. For the benchmark we took the original dataset and repurposed it in TACO format with no modifications.

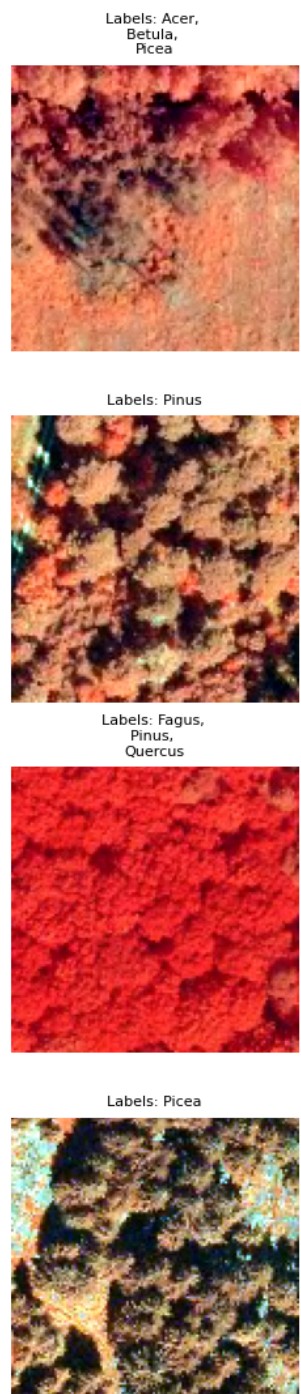

Figure 18: Training Set Examples for TreeSatAI dataset.

## C.16 Substation

This dataset was curated by TransitionZero and sourced from publicly available data repositories, including OpenStreetMap and Copernicus Sentinel data (Jindgar & Lindsay, 2024). The dataset consists of Sentinel-2 228x228 pixel images from 27k+ global locations; the original task was to semantically segment power substations. For the benchmark, we converted the original labels into instance segmentation labels. Furthermore,

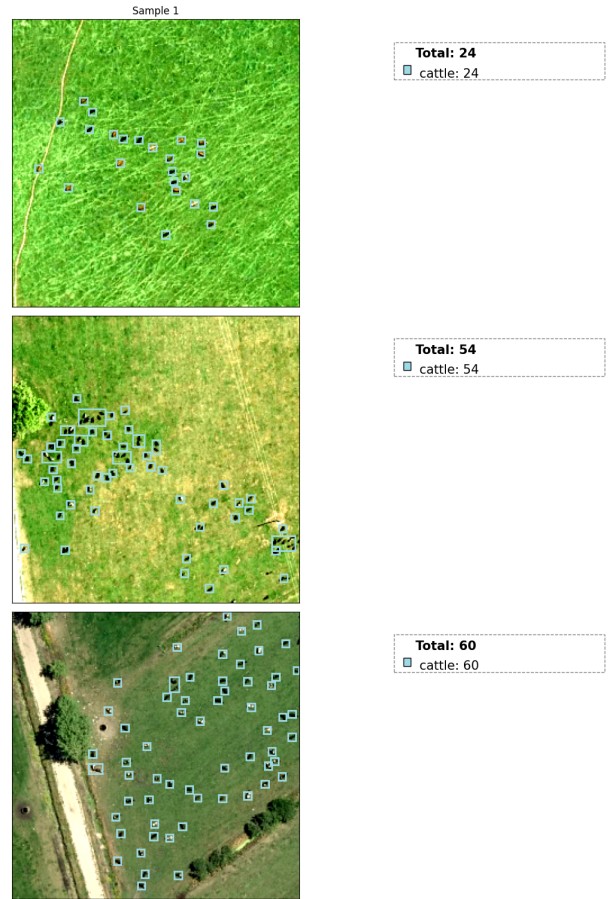

Figure 19: Training Set Examples for NZCattle dataset.

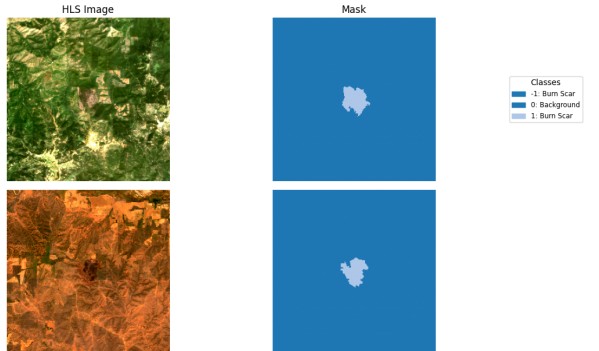

Figure 20: Training Set Examples for BurnScars dataset.

as most locations had 4-5 images taken at different timepoints (i.e., revisits), but beyond cloud occlusions, there is, in principle, no need for multiple timestamps to complete the task, we created a cloud-free composite for each location. Data was sub-sampled for each split to reduce computational requirements.

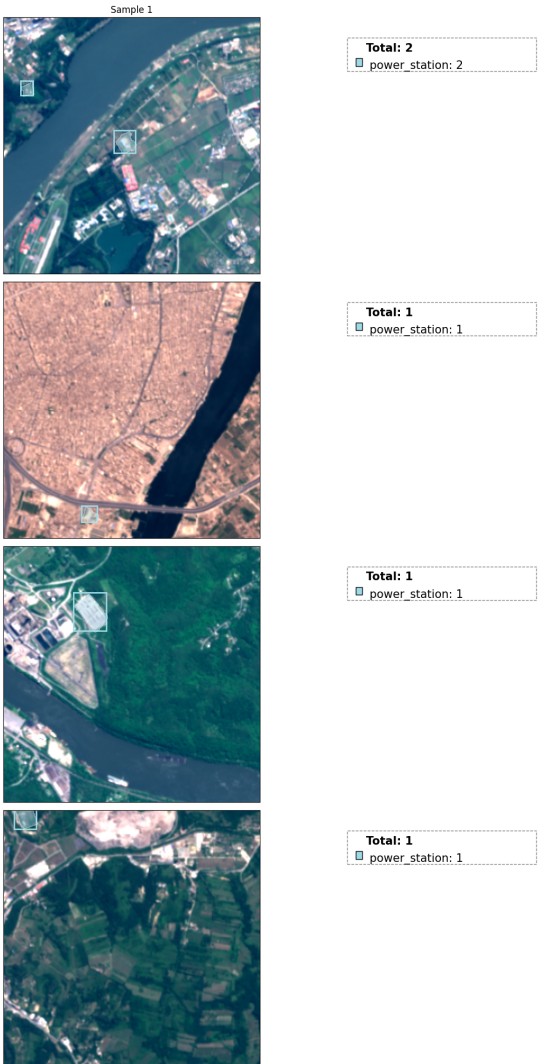

Figure 21: Training Set Examples for Substation dataset.

### C.17 So2Sat

So2Sat is a dataset designed for multimodal classification. The aim is to classify local Climate Zones globally. We downloaded the modified version of this dataset from GEO-Bench (Lacoste et al., 2023) and repurposed it in TACO format with no modifications.

### C.18 Forestnet

Forestnet is a curated dataset of Landsat 8 satellite images of known forest loss events paired with driver annotations from expert interpreters in South Asia. We downloaded the modified version of this dataset from GEO-Bench (Lacoste et al., 2023) and repurposed it in TACO format with no modifications.

### C.19 Dataset Geographical Distribution by continent

Figure 24 shows the geographical distribution of dataset samples across continents.

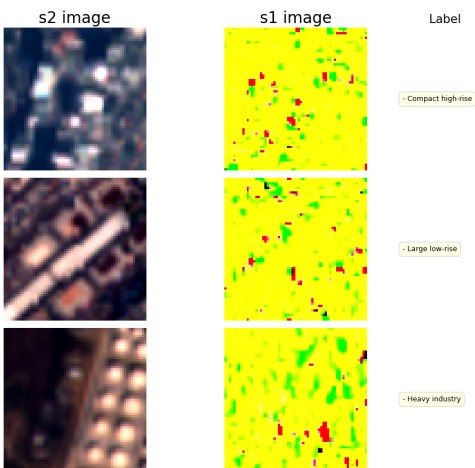

Figure 22: Training Set Examples for So2Sat dataset.

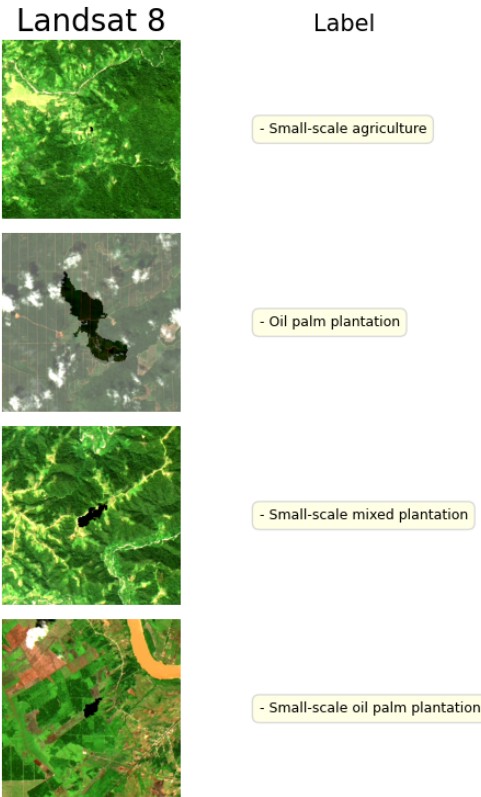

Figure 23: Training Set Examples for Forestnet dataset.

## C.20  Dataset Statistics

Z-score normalization statistics were computed modality and band-wise for each dataset on the training set split and subsequently used to normalize all inputs. For the BioMassters dataset that contains continuous pixel-wise regression labels, z-score normalization was also used for the labels.

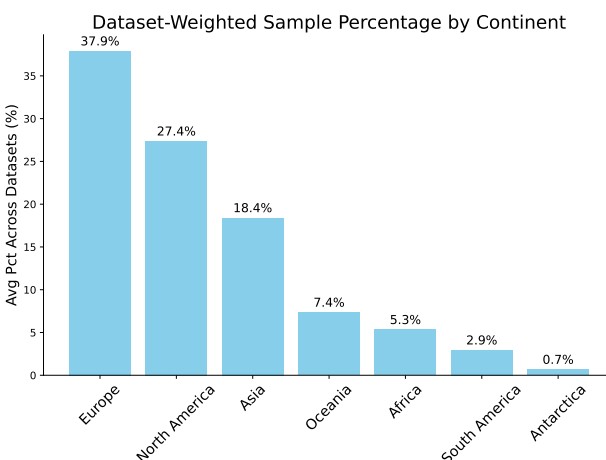

Figure 24: Distribution of samples weighted by source dataset size across continents. Each dataset's continent distribution is normalized by its dataset size before aggregation.

# D Detailed experimental setup

To ensure reliable results, the evaluation protocol employs a combination of hyperparameter optimization (HPO) followed by repeated seed experiments. For segmentation, regression, and classification tasks, we conduct 16 trials during the HPO phase. Due to higher computational requirements, detection experiments use 10 trials. After the HPO process for a given model and dataset combination, the set of parameters yielding the best validation set metric is selected and subsequently used for repeated experiments with five different random seeds. The final performance is then calculated on the test for each seed. Both HPO and repeated experiments are configured through the TerraTorch Iterate library, which automates the benchmarking workflow. HPO employs Optuna (Akiba et al., 2019) with Bayesian optimization and a Tree Structured Parzen Estimator (TPE) for efficient sampling. The general optimization search space was fixed across all models and datasets:

- Learning Rate: A real value ranging from 1e-6 to 1e-3.

- Batch Size: An integer value selected from the list: $\{8, 16, 32\}$.

An AdamW optimizer is used with a fixed weight decay of 0.01. Training incorporates early stopping with a patience of 10 epochs. For object detection, the batch size is fixed at 8, and HPO is conducted solely on the learning rate.

All experiments in this project were executed using a single GPU across two primary computing clusters. Depending on availability, either NVIDIA V100 (32 GB) or NVIDIA A100 (80 GB) GPUs were used opportunistically. Over the course of the entire project, including ablation studies, a total of approximately 68,000 GPU hours were consumed, with around 50,000 GPU hours devoted to hyperparameter optimization and 18,000 GPU hours to repeated experiments. Because the experiments were conducted on heterogeneous systems and GPU architectures, these numbers should be regarded as approximate and are intended to provide a high-level overview of the project's computational resource utilization.

## D.1 Main Raw Results by Dataset

The raw results obtained using these base experimental settings for each of the 19 datasets are presented in Figure 25. Beyond the main experimental settings, the succeeding sections give results from the various ablations conducted to understand the effect of changing settings on benchmarking.

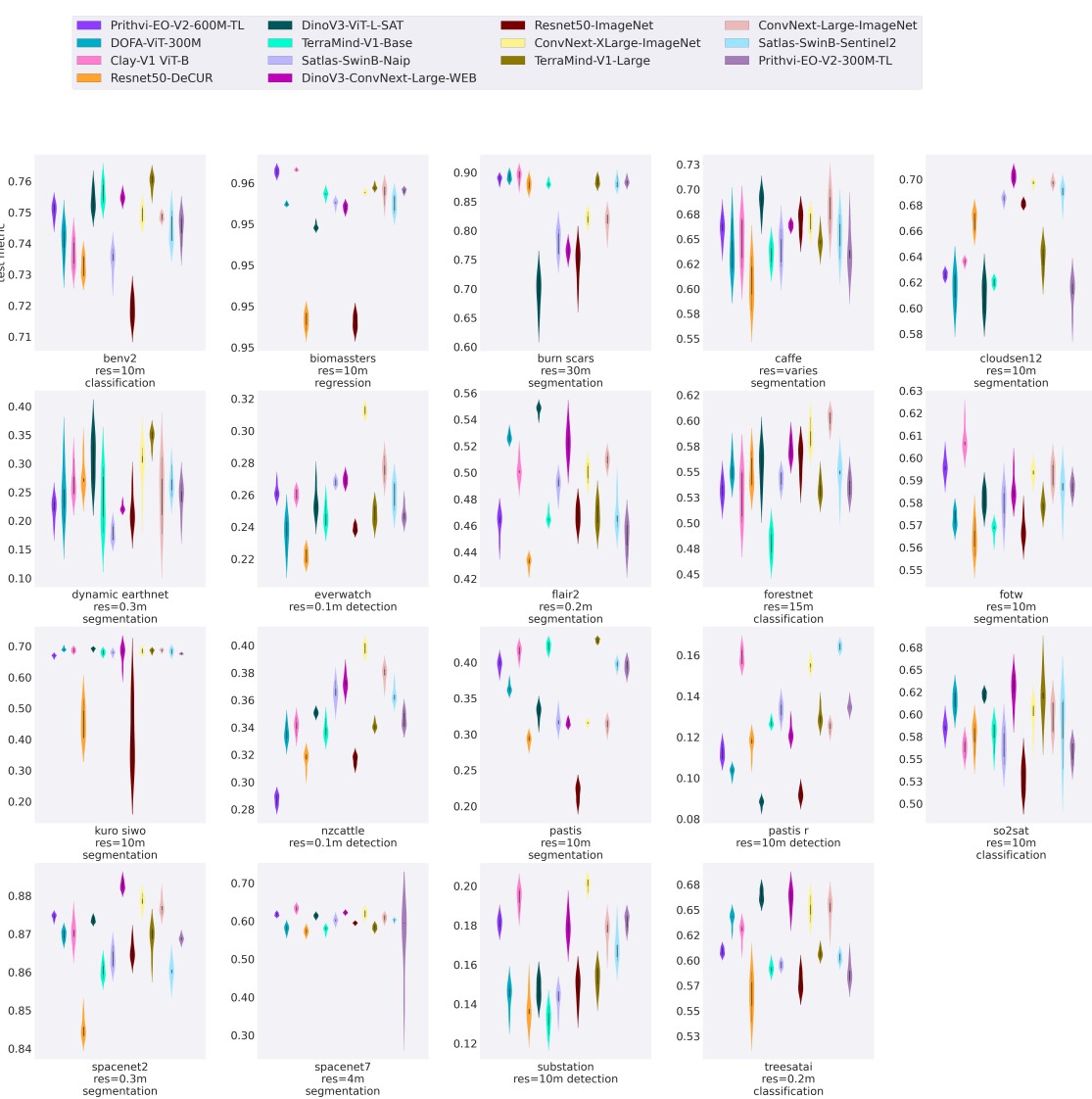

Figure 25: Main results: Raw performance by dataset.

# E    Ablations

## E.1    Frozen Encoder Ablation

To assess the impact of freezing the encoder during training, we performed an ablation where the encoder remains fixed while all other settings match the main experiments. This ablation was applied to all pixel-wise and classification datasets using a subset of models. Figure 26 shows a side-by-side comparison of full fine-tuning versus frozen encoder aggregated performance for each model, while Figure 27 presents the raw results per dataset. A systematic drop in performance is observed when the encoder is frozen, suggesting that all models benefit from some fine-tuning. Importantly, a change in model ranking is also observed. Figure 4 shows the aggregate change in model rank due to this ablation along with rank changes due to other ablations.

It should be noted that the drop in performance is not the same across all datasets. To investigate the effect size by dataset, a paired t-test is performed and the Cohen's D effect size is computed for datasets that have a statistically significant change. Figure 28 shows the Cohen's D effect size for datasets with a statistically

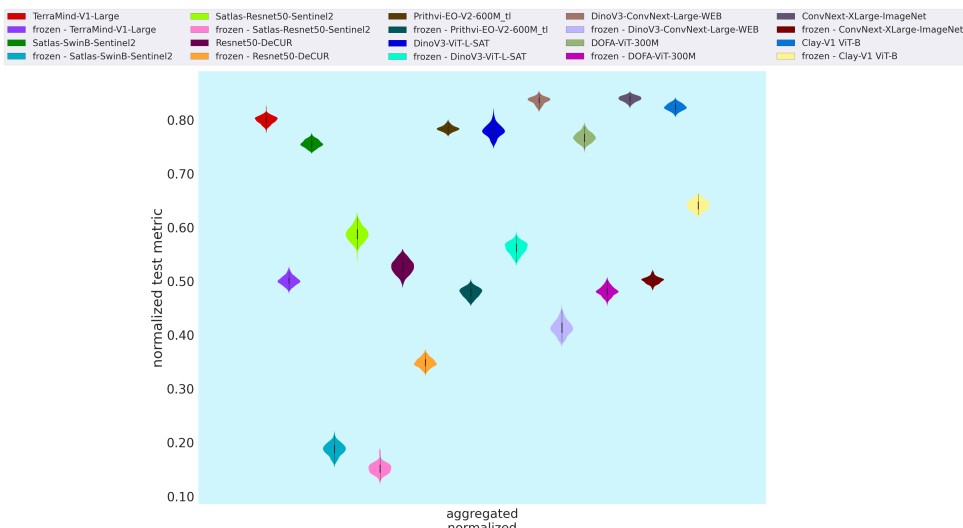

Figure 26: Aggregated Frozen Vs Unfrozen Ablation: freezing the encoder results in a systematic drop in performance.

significant drop in test metric due to frozen backbones (aggregated across all models). Datasets such as treesatai and benv2 have the biggest drop in performance with encoder freezing.

## E.2    Linear Decoder Ablation

To evaluate the impact of using a linear decoder for pixel-wise tasks, we ran an ablation replacing the UNet decoder with a single linear layer, keeping all other settings identical to the main experiments. This was applied to all pixel-wise datasets (segmentation and regression) using a subset of models.

Figure 29 compares aggregated performance of UNet versus Linear decoder for each model, while Figure 30 shows the raw results per dataset. Again, a linear decoder shows lower performance and substantially affects scoring. Figure 31 shows variation in Cohen's D effect size per dataset for datasets with statistically significant reduction. In particular, fotw and spacenet7 datasets have the largest decrease. Figure 4 demonstrates that while there is an aggregate change in model rank due to this ablation, the difference in rank is smaller than changes observed due to other ablations.

## E.3    Multi-Spectral Vs RGB Data Ablation

To assess the role of multi-spectral data, we ran an ablation using only GeoFM encoders on datasets with additional bands beyond RGB. Unlike the main experiments, which used all available bands, this ablation restricted inputs to RGB only, while keeping all other settings unchanged.

Figure 33 shows the raw per-dataset results, while 32 shows aggregated performance. A general drop in performance is observed in both figures when multi-spectral bands are excluded, thus highlighting the importance of multi-spectral data. Similar to the other ablations, the Cohen's D effect size is computed for datasets that have a statistically significant change according to a paired t-test. Datasets with a statistically significant drop in performance due to excluding multi-spectral bands (aggregated across all models) are highlighted in Figure 34. The burn scar dataset suffers the biggest reduction by far, possibly due to some burn scars not being detectable with simple RGB data. This emphasizes the importance of multi-spectral data for some downstream tasks in EO. The Multi-Spectral-dependent capability is comprised of datasets with a statistically significant change.

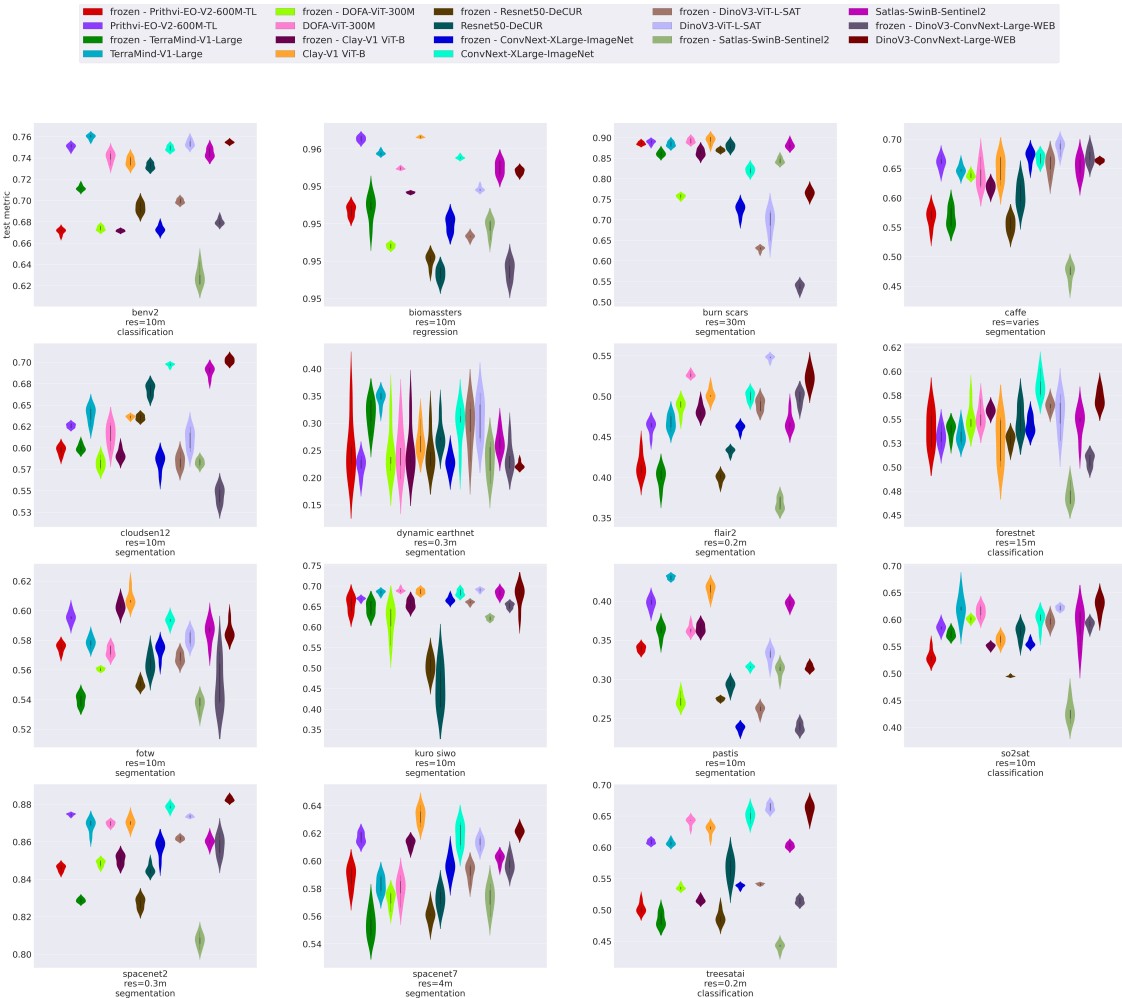

Figure 27: Frozen Vs Unfrozen Encoder Ablation: Raw results per dataset.

## E.4 Ground Sampling Distance (GSD) Ablation

This ablation applies a 14x14 kernel filter to increase the Ground Sampling Distance (GSD) of validation and test images, while train images are kept at the original resolution. The best HPO settings are used for training with 5 repeated seeds for each model on each dataset.

All other settings used in the main experiments were maintained. Datasets with a resolution under 10m are used with a subset of models. Ground sampling distance is an important factor in satellite data, determining the level of detail that can be observed in a satellite image. A lower GSD (high resolution) implies that each pixel covers a smaller area on the ground and finer details are clearer when looking at the overall image. Figure 35 shows raw results per dataset from this ablation, while 36 shows aggregated performance. Both figures highlight that increasing GSD tends to lower test metrics across all models. This is expected as the images with lower (original) GSD have more details. However, it is notable that some datasets exhibit a smaller effect size as demonstrated by the results Cohen's D effect sizes of Figure 37.

## E.5 Multi-Modality Ablation

To investigate the effect of multi-modal inputs, this ablation applies both optical and SAR data simultaneously to each model versus only using optical data (e.g. Sentinel 2 or Planet data). All other settings used in

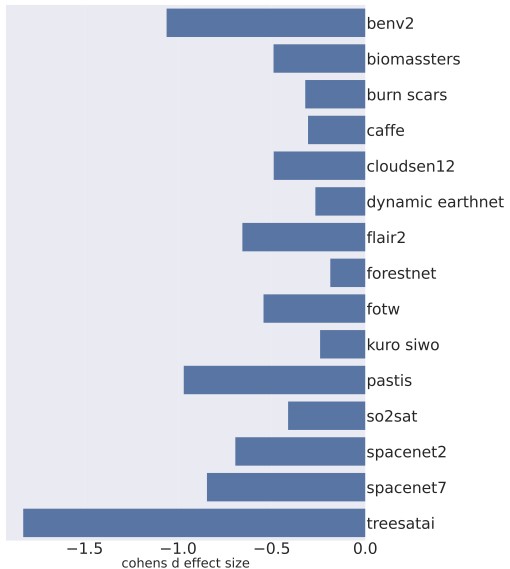

Figure 28: Cohen's D effect size for Unfrozen/Frozen Ablation: Per dataset reduction in performance due to freezing encoder

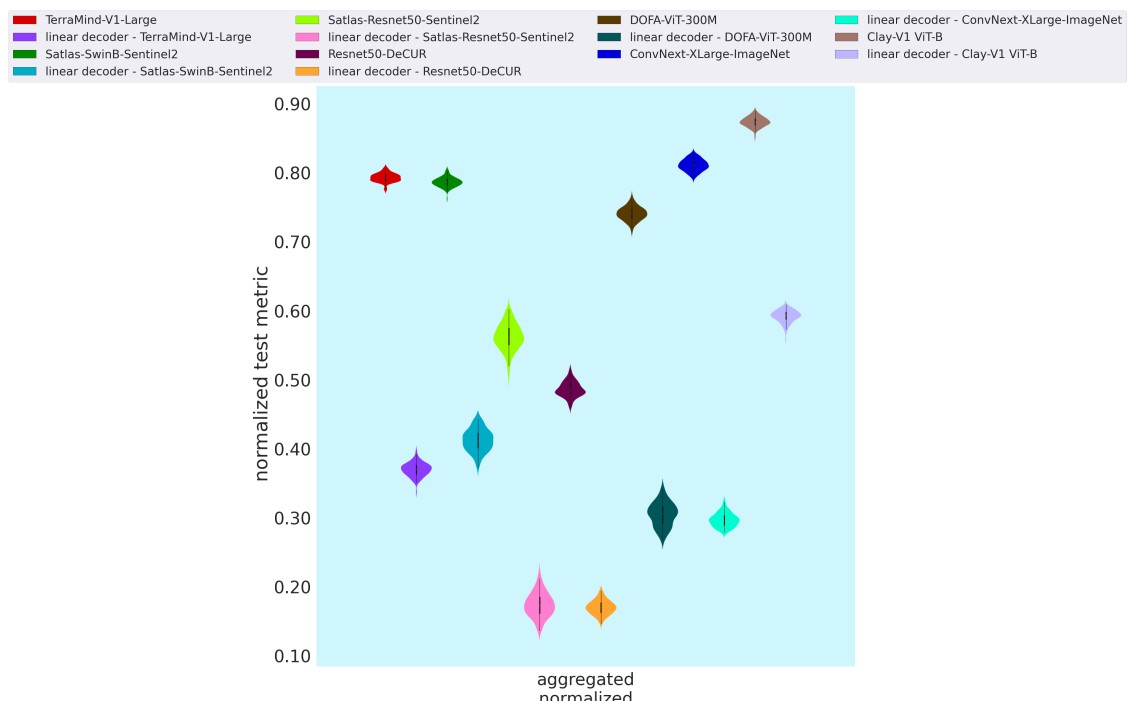

Figure 29: Aggregated UNet Vs Linear Decoder Ablation: linear decoder lowers performance for all models.

the main experiments were maintained. The raw results of Figure 38 show that this ablation is particularly dependent on the dataset and model.

For the pastis dataset, multi-modal inputs reduce test metrics regardless of the model, suggesting that the combination of the multi-modal inputs may not be beneficial for this dataset. Conversely, the biomassters dataset consistently shows an improvement across models. BigEarthNet V2 dataset has more nuanced results: while TerraMind-V1-Large increases test metrics with multi-modal inputs, the remaining models have either

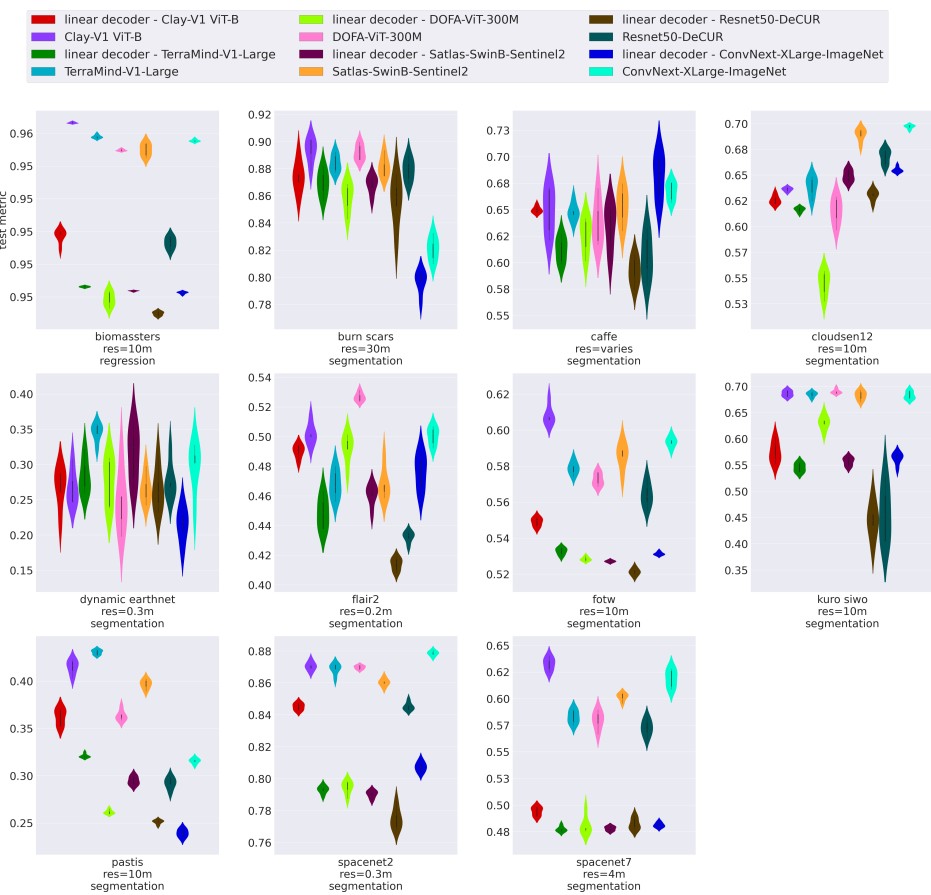

Figure 30: UNet Vs Linear Decoder Ablation: Raw results per dataset.

similar or lower performance. Overall, the effect of multi-modal inputs is inconclusive as performance appears to vary by model and dataset.

### E.6 Multi-Temporal Vs Mono-Temporal Ablation

To evaluate the effect of using a single timestamp, we ran an ablation on multi-temporal datasets where inputs were restricted to one timestamp instead of multiple (up to seven in the main experiments), keeping all other settings unchanged.

Applied to a subset of models, this ablation revealed substantial performance drops across some datasets such as Pastis, thus confirming the importance of temporal information for these particular datasets in the multi-temporal capability. Raw per-dataset results are shown in Figure 39. Additionally, Figure 4 also shows that changing from multi-temporal to mono-temporal inputs causes the biggest change in rank, second only to the effect of freezing the encoder.

### E.7 Cosine Annealing Ablation

In this ablation, we applied a cosine annealing schedule to adjust the learning rate during training. Figure 40 shows the raw per-dataset results. Outcomes varied across datasets and models, providing no definitive conclusion.

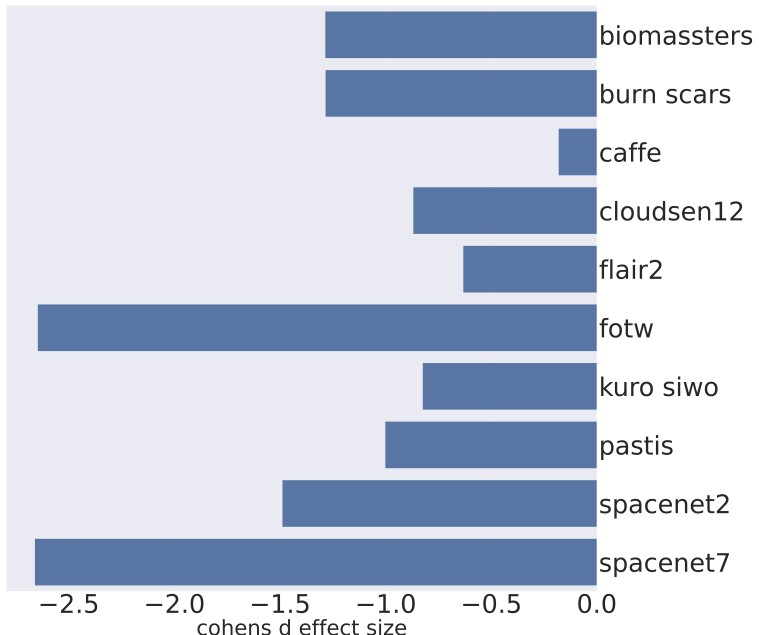

Figure 31: Cohen's D effect size for UNet/Linear Decoder Ablation: Per dataset reduction in performance due to linear decoder.

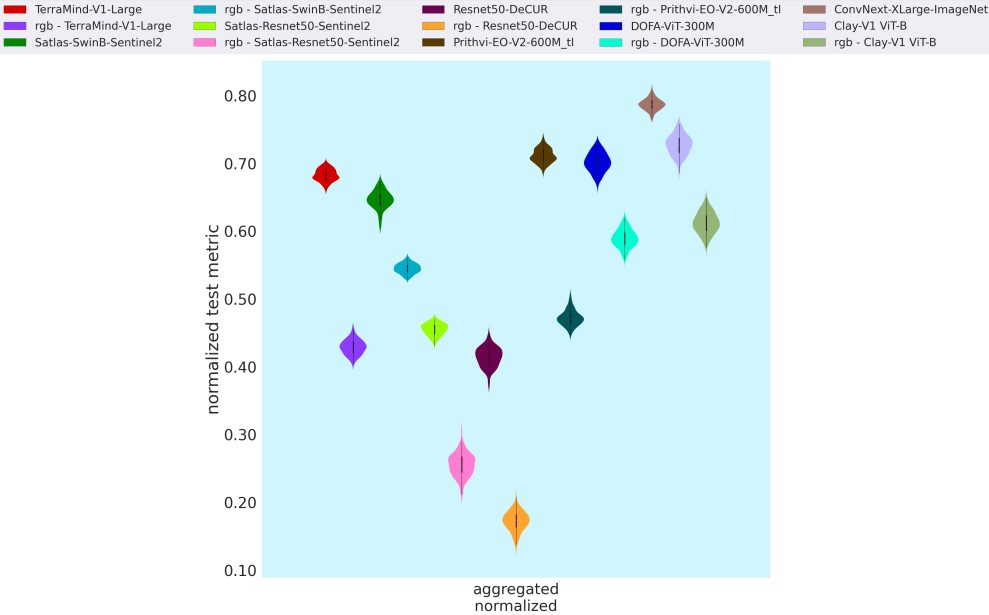

Figure 32: Aggregated Multi-Spectral Vs RGB Data Ablation: excluding multi-spectral data tends to reduce performance.

## E.8 Linear Warm Up Ablation

In this ablation, we applied linear warm-up to adjust the learning rate during the first five training epochs. Figure 41 shows the raw per-dataset results. Outcomes varied across datasets and models not showing any clear trend.

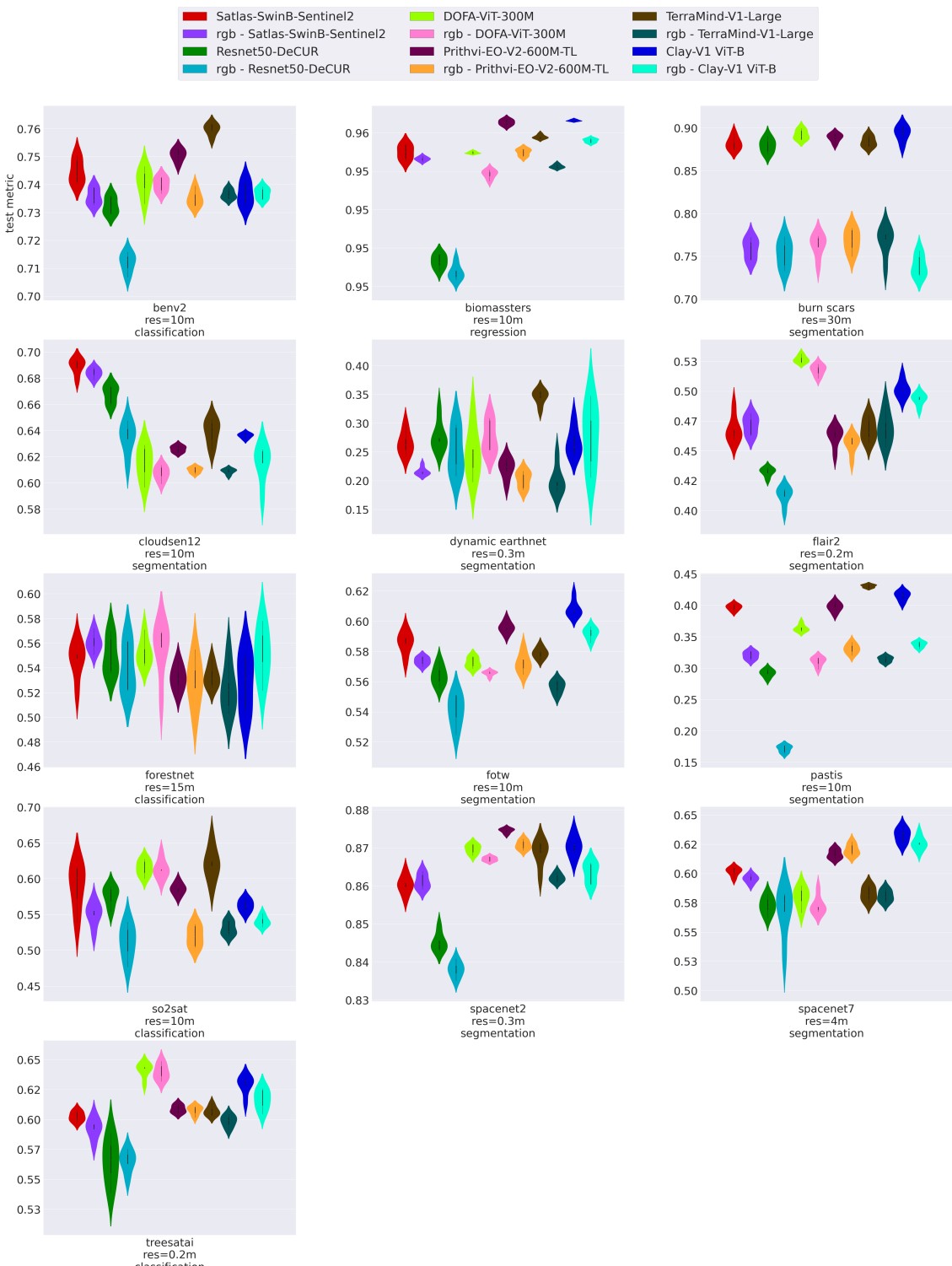

Figure 33: Multi-Spectral Vs RGB Data Ablation: Raw results per dataset.

### E.9 Leave-One-Out Robustness Analysis

**Multi-Spectral-Dependent Capability.** To assess whether the Multi-Spectral-Dependent capability ranking is disproportionately influenced by any single dataset, we conducted a leave-one-out robustness

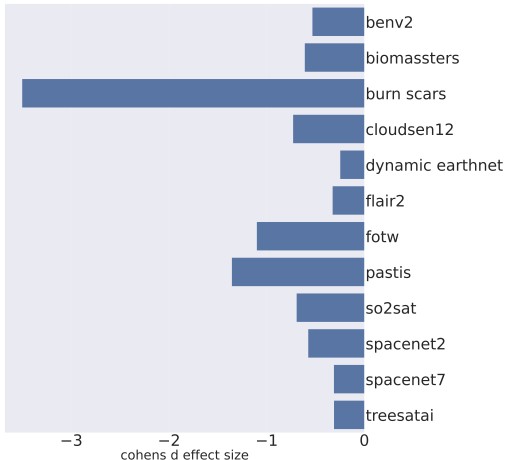

Figure 34: Cohen's D effect size for Multi-Spectral/RGB Ablation: Per dataset reduction in performance due to excluding multi-spectral data.

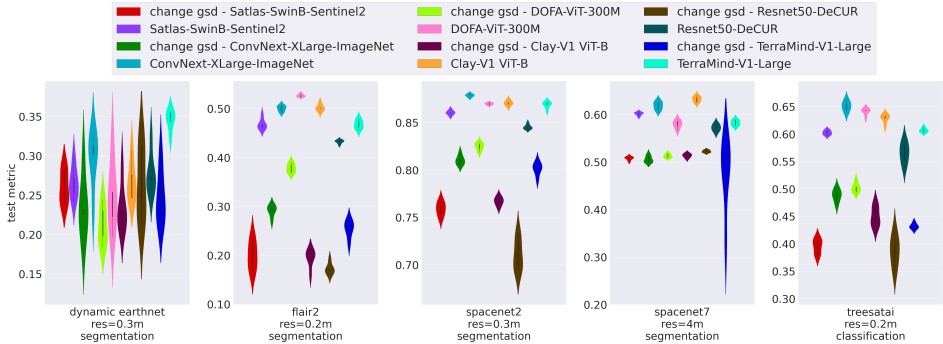

Figure 35: Increasing Ground Sampling Distance Ablation: Raw results on 5 datasets.

analysis in which the capability ranking was recomputed after excluding each multi-spectral dataset in turn. Figure 42 visualises the resulting model ranks across all leave-one-out settings. Consistent with the per-dataset effect sizes reported in Appendix E.3, `burn_scars` exhibits a strong influence on absolute performance due to its high spectral sensitivity; however, its removal does not materially affect the overall structure of the capability ranking. In particular, the top-ranked and bottom-ranked models remain unchanged across all leave-one-out conditions. For the majority of models, rank variations are limited to shifts of one or two positions, indicating that the Multi-Spectral-Dependent ranking is not solely driven by a single dataset. Larger rank changes are observed for a small subset of models when specific datasets are removed (e.g., Satlas-Swin-100M upon exclusion of CloudSen12), highlighting model-specific dependencies on particular datasets. Overall, this analysis demonstrates that while individual datasets contribute unevenly to the aggregate score, the Multi-Spectral-Dependent capability ranking is stable at the group level and reflects a consistent signal across datasets rather than a single-dataset effect.

**Multi-Temporal Capability.** We performed a leave-one-out analysis to assess whether the Multi-Temporal capability ranking is driven by any single dataset, recomputing the ranking after excluding each temporal benchmark in turn. Figure 43 shows that the ranking is highly stable under dataset removal: the top-ranked and bottom-ranked models remain unchanged across all settings, and most models shift by at most one or two positions. Larger rank changes occur for a small number of models when specific datasets (e.g., PASTIS-R) are excluded, reflecting model-specific dependencies rather than dominance of a single dataset. Overall, this analysis indicates that the Multi-Temporal capability captures a consistent temporal performance signal across datasets.

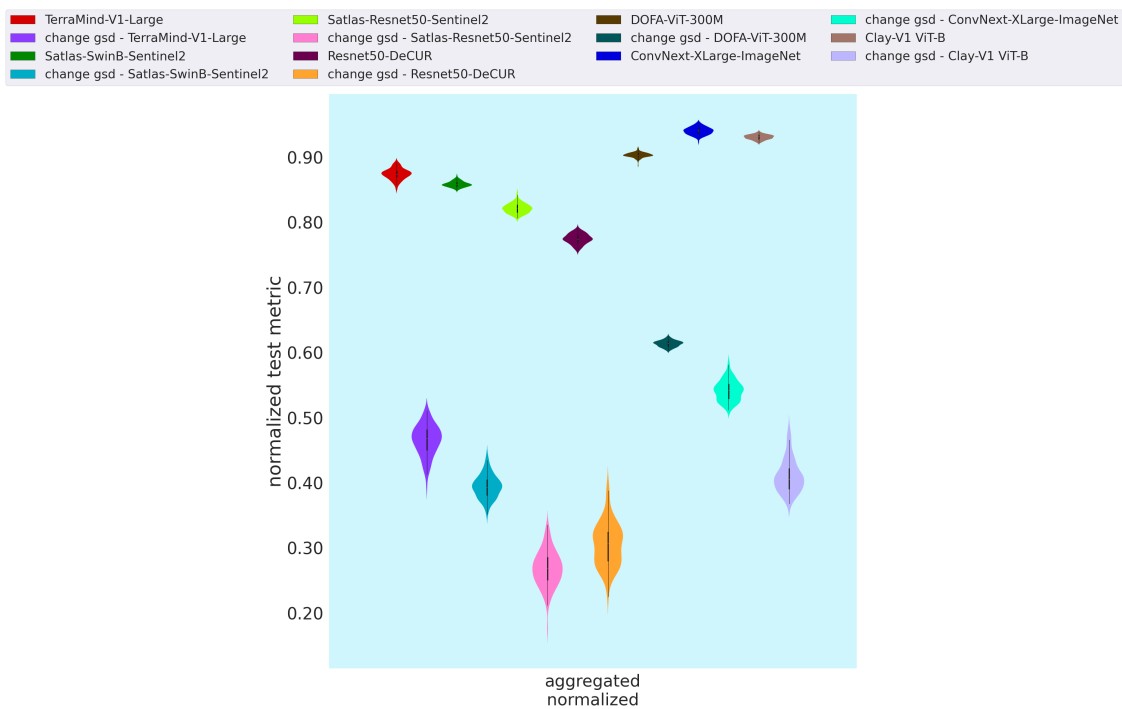

Figure 36: Aggregated Original GSD Vs Increased GSD: lower GSD can improve performance in some cases.

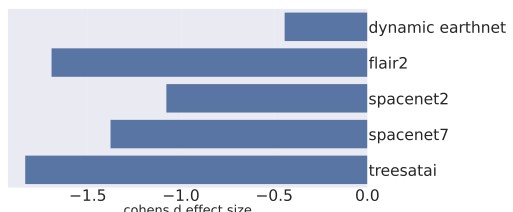

Figure 37: Cohen's D effect size for Increasing Ground Sampling Distance Ablation: Per dataset reduction in performance due to increased GSD.

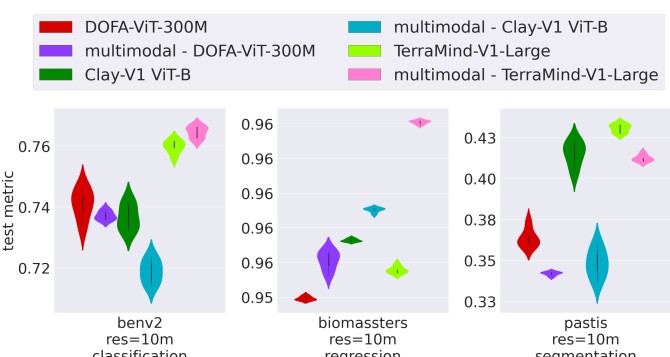

Figure 38: Multi-modal Vs Optical only: change in performance with multi-modal input vs Optical input only. Ablation on all multi-modal datasets and models only

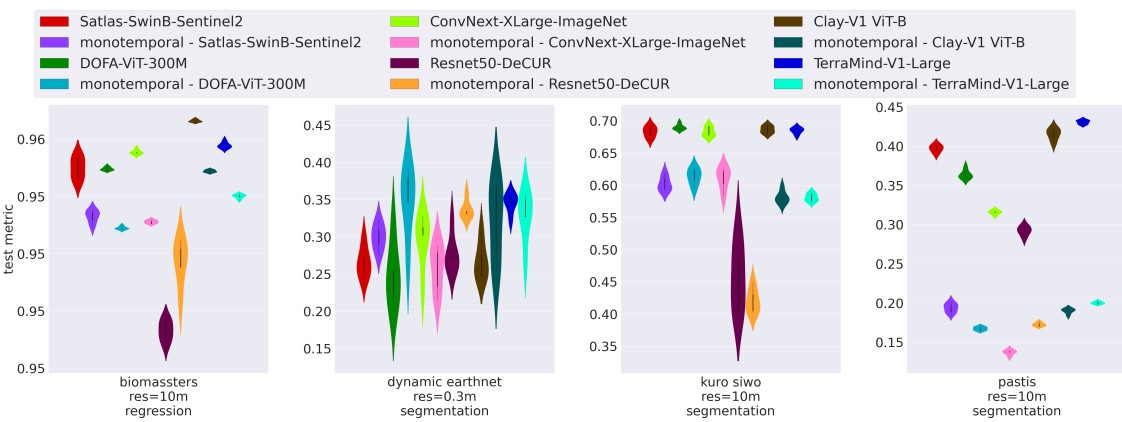

Figure 39: Multi-Temporal Vs Mono-Temporal Ablation: raw results on 4 datasets.

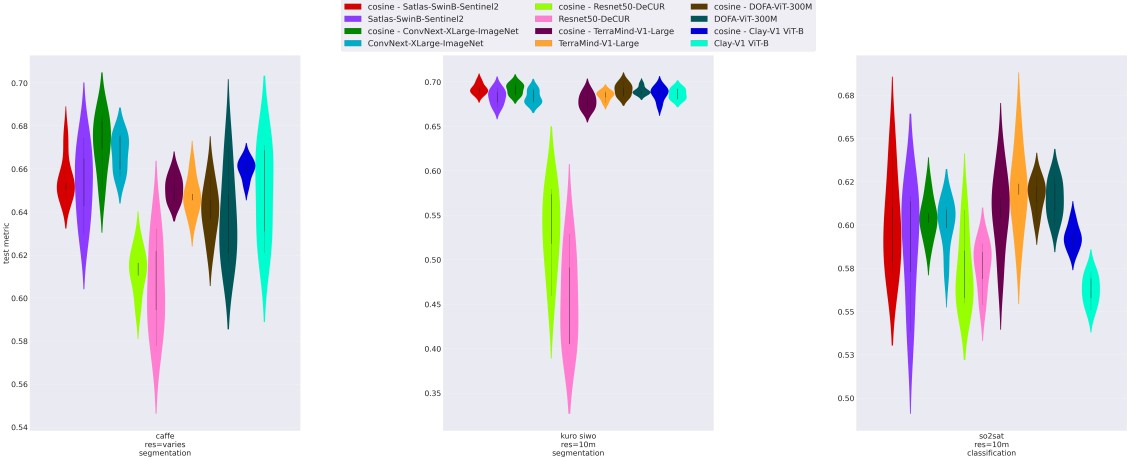

Figure 40: Cosine Annealing Learning Rate vs. Fixed Learning Rate: raw results on 3 datasets are inconclusive.

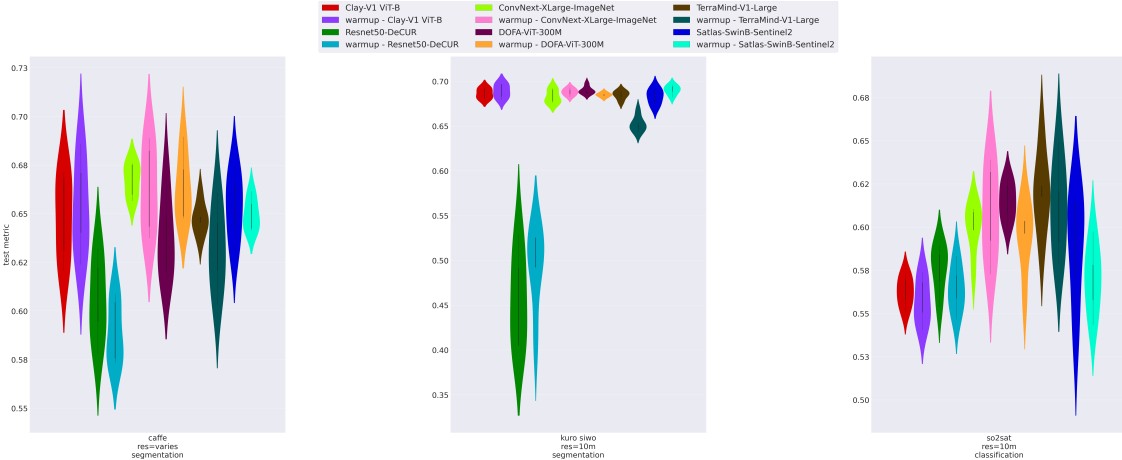

Figure 41: Linear Warm-Up vs. Fixed Learning Rate: raw results on 3 datasets are inconclusive.

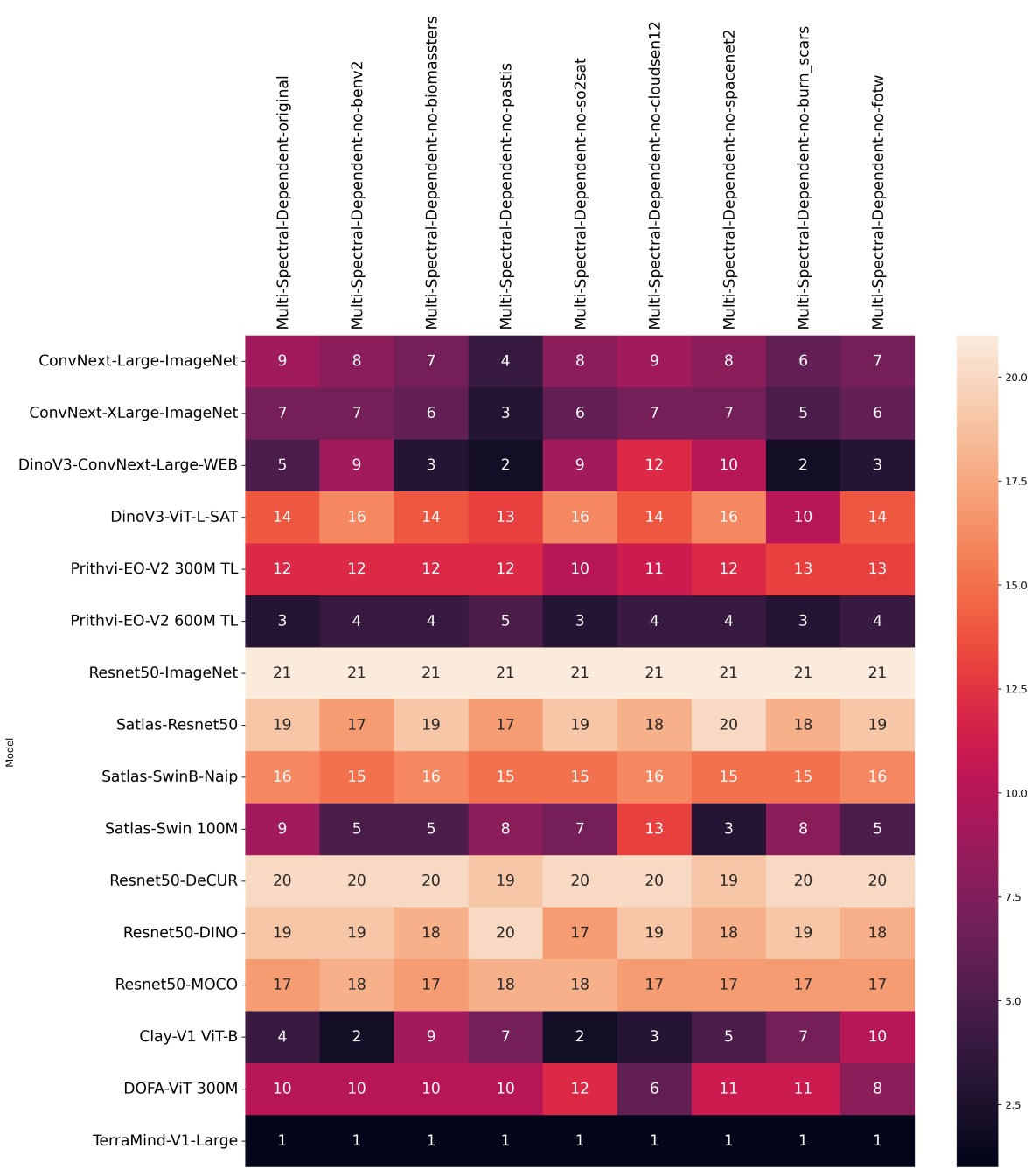

Figure 42: Leave-one-out robustness analysis for the Multi-Spectral-Dependent capability. Each column shows model ranks when one dataset is excluded. The ranking structure remains stable across all leave-one-out conditions.

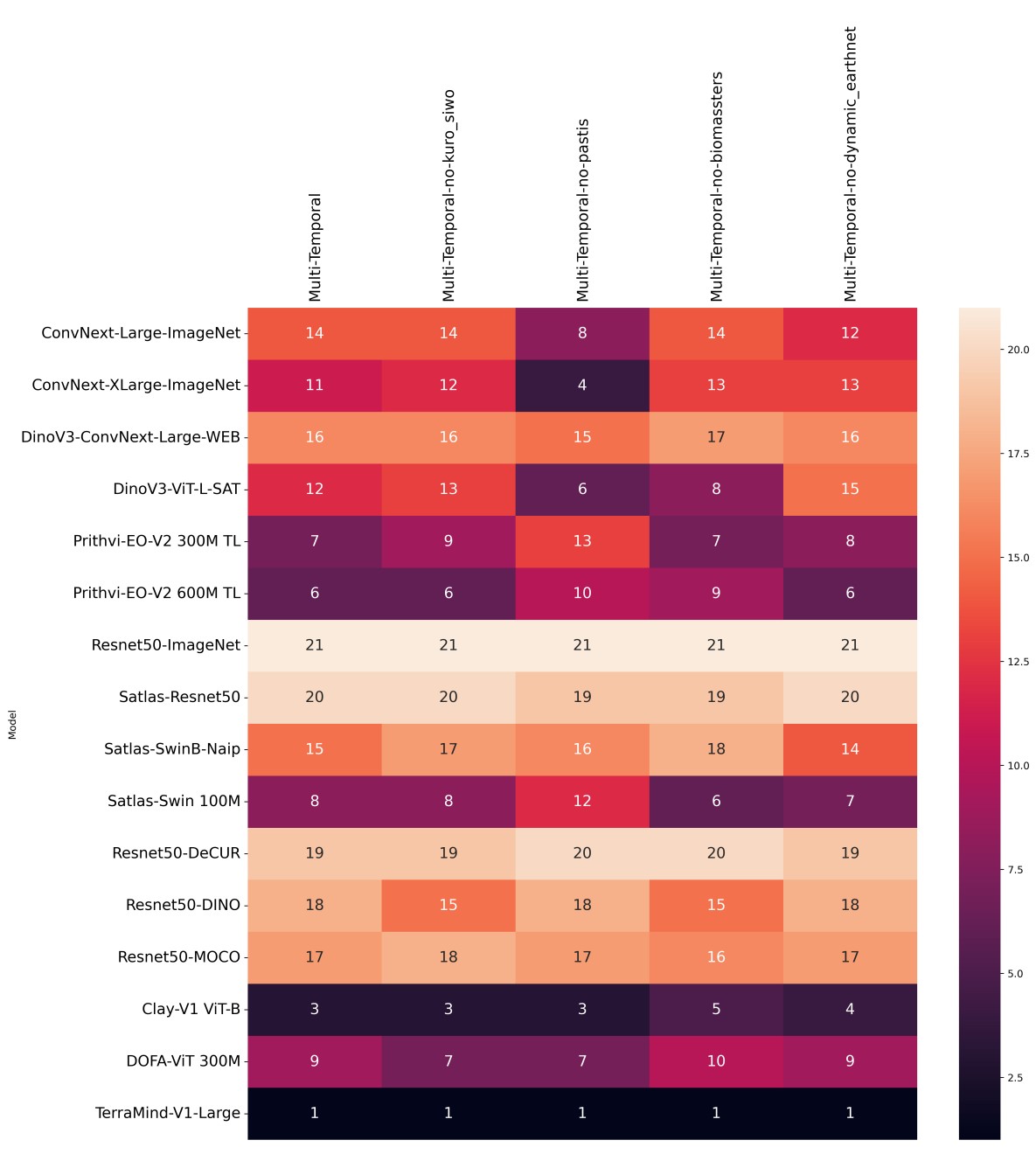

Figure 43: Leave-one-out robustness analysis for the Multi-Temporal capability. Each column shows model ranks when one dataset is excluded. The ranking remains highly stable across all leave-one-out conditions.

