# OpenReview forum: "GEO-Bench-2: From Performance to Capability,  Rethinking Evaluation in Geospatial AI"
_TMLR — Decision pending for TMLR_

### Review · Reviewer_4cCk · 2026-03-24

**Summary Of Contributions:**

This paper addresses the absence of standardized, multi-dimensional evaluation for GeoFMs. The authors argue that existing benchmarks collapse model quality into a single aggregate score, masking capability-specific strengths and weaknesses that matter for real deployment decisions.

They propose a benchmark organized around nine overlapping capability groups, defined by task type, input modality, spatial resolution, and temporal structure. These are evaluated across multiple models on 19 EO datasets covering classification, segmentation, regression, detection, and instance segmentation. The evaluation protocol is standardized (HPO, augmentation, decoder design, IQM-based aggregation) and is backed by a public leaderboard and TerraTorch integration.

The central finding is that no single model dominates across all tasks, and that EO-specific models hold a measurable advantage on multi-spectral tasks while large natural-image models are competitive on RGB tasks. The paper also includes substantial ablation studies on encoder freezing, decoder choice, and temporal modeling, with the temporal modeling ablation producing the largest rank perturbation outside of encoder freezing.

**Additional Comments:**

The submission reflects significant effort and the experimental scope is impressive. The capability-group framing is a clean and well-motivated contribution, and the ablation studies, particularly the frozen encoder and multi-temporal ablations, are informative for the field. The decision to recommend full fine-tuning as the primary evaluation setting is well-justified by the results.

The discussion in Section 5 is honest about limitations and avoids overclaiming. The observation that EO has not yet exhibited LLM-style scaling laws is an interesting point that could be developed further in future work. The main issues standing between this work and acceptance are outlined in the Requested Changes section above. I think these are all addressable in a focused revision.

**Audience:**

Yes

**Audience Explanation:**

The paper's capability-group framing offers a useful conceptual lens for model selection. The findings, especially the ablation studies, are practically important and will benefit the remote sensing and EO ML community. The leaderboard infrastructure further extends the paper's value beyond its initial experiments.

**Broader Impact Concerns:**

The paper's broader impact statement is appropriate and honest about geographic bias (Eurocentric data). The work serves a clarifying and organizational role and does not raise significant broader-impact concerns.

**Claims And Evidence:**

No

**Claims Explanation:**

The central claim that GeoFM evaluation is inherently multi-dimensional and that aggregate scores obscure meaningful capability differences is well-supported. The ranking shifts across capability groups in Tables 1 and 5 are large and consistent, and the ablations directly validate the secondary claims about frozen backbones, decoder design, and multi-temporal inputs.

That said, several specific claims rest on design choices whose implications for certain models are not fully discussed. The "no single model dominates" framing also needs stronger statistical grounding, as several scores in Figure 3 are close enough that significance is unclear. These concerns are outlined in the Requested Changes section below.

**Requested Changes:**

The changes below are aimed at strengthening the evidential basis for the main claims and improving the paper's completeness and accessibility.

### Content

- [C1] The "Core" capability is the primary ranking reported in Table 1 and Figure 3, yet the selection criteria for which datasets are included are never stated. Section 2.2 describes Core only as "a balanced subset across tasks, resolutions, and modalities," which seems insufficient. The authors should clarify this selection in Section 2.2 or a dedicated appendix, ideally using the same discriminative-power criterion applied to the full dataset selection in Section 2.1.

- [C2] Section 3.5 processes each timestamp independently and applies uniform average-pooling before the decoder for all models. This is a reasonable choice for isolating backbone quality in a controlled setting. However, since the benchmark defines a "Multi-Temporal" capability group, it is worth clarifying whether this protocol is intended to measure temporal modeling ability or backbone expressiveness on temporal data since these are different things. Models with native temporal heads might not be able to leverage that capacity under this setup. A brief discussion of this design choice and its implications for interpreting the Multi-Temporal capability rankings would strengthen the paper, particularly given that Section 4.6 identifies temporal modeling as "an underexploited yet decisive factor."

- [C3] Figure 3 reports bootstrapped IQM scores with confidence intervals, but does not include pairwise significance tests between models. For closely ranked models, overlapping CIs make it difficult to assess whether the differences are meaningful. The ablations in Appendix D.1 and D.3 already use paired t-tests for this purpose. Applying a similar approach to the capability-level comparisons in Figure 3 would help ground the ranking claims.

- [C4] The multi-spectral ablation (Appendix D.3) shows that burn_scars has a substantially larger Cohen's D effect size than all other datasets. Section D.3 itself notes it "suffers the biggest reduction by far." This raises a question about whether the Multi-Spectral-Dependent capability ranking is broadly representative or largely driven by one dataset's sensitivity to band exclusion. Reporting per-dataset effect sizes alongside the aggregate score and briefly discussing this concentration would help readers assess the generality of the multi-spectral importance claim. It is also worth noting that Appendix D.5 finds the multi-modal (SAR + optical) ablation inconclusive across models, which adds to the uncertainty around what the Multi-Spectral-Dependent capability is measuring.

- [C5] A similar observation applies to the Multi-Temporal capability. Figure 39 shows that the performance gap between multi-temporal and mono-temporal inputs is most pronounced for PASTIS, with BioMasters, DynamicEarthNet, and KuroSiwo showing smaller effects. A brief discussion of whether the capability ranking is robust across datasets, or primarily shaped by PASTIS, would strengthen the interpretation of this capability group.

- [C6] (good to have) The paper does not report the total compute used to produce the benchmark results. Even a rough figure (e.g., total GPU-hours on specific hardware) in Section 3.1 or Appendix C would improve reproducibility and completeness.

- [C7] (good to have) The paper does not discuss versioning or dataset deprecation. A brief note on how the benchmark would handle dataset replacements, normalization anchor updates, and historical comparability would be a useful addition.

- [C8] (good to have) The paper positions itself as offering a more informative evaluation than prior benchmarks such as PANGAEA and GEO-Bench, but does not include a direct comparison of model rankings across benchmarks. A side-by-side comparison on overlapping models and datasets would provide concrete evidence that the capability-group framing surfaces meaningfully different conclusions.

### Editorial

- [E1] The paper does not include a dedicated related work section. Prior benchmarks (PANGAEA, GEO-Bench, CopernicusBench, PhileoBench) are mentioned in the introduction but not systematically compared in terms of design choices, dataset coverage, or evaluation protocols. Adding a related work section would help readers better contextualize the contributions.

- [E2] The reference list has several inconsistencies. Section 3.3 cites Girshick (2015) for Faster R-CNN, but Girshick (2015) is Fast R-CNN; the correct citation is Ren et al. (2015, NeurIPS). Additionally, several references appear twice under a/b suffixes but refer to the same work: Ahlswede et al. 2022, Bountos et al. 2024, Clasen et al. 2024, Gourmelon et al. 2022, Kerner et al. 2025, Nascetti et al. 2023, and Toker et al. 2022. These should be consolidated.

- [E3] (good to have) The paper is tightly integrated with TerraTorch, using it for HPO through TerraTorch Iterate and baseline decoders via TerraTorch components. This may create friction for researchers working with custom training pipelines. A brief note clarifying which benchmark requirements are framework-agnostic (e.g., dataset splits, metrics, aggregation methodology) versus TerraTorch-specific would improve accessibility.

Addressing C1–C5 is necessary before the primary claims can be considered well-supported. E1–E2 are required for completeness and correctness. The remaining items (C6–C8, E3) would further strengthen the submission but are not blocking.

---

> ### Author Response · Authors · 2026-04-22
> **Response to Reviewer - Content**
>
> We thank the reviewer for the thorough and constructive feedback.
>
> **C1**: We agree that the selection criteria for the Core capability were not sufficiently described. The Core capability is intended to provide a computationally efficient yet discriminative subset that spans all major challenges of the benchmark. We have expanded Section 2.2 to make this explicit: Core datasets were selected by applying the same discriminative-power criterion from Section 2.1, with the additional constraint that no single task type, resolution range, or spectral profile is over-represented. This ensures aggregate Core scores reflect broad model competence rather than strength in any one dimension.
>
> **C2**: The reviewer correctly identified a tension between the Multi-Temporal capability's intent and the protocol as written. Average-pooling of per-timestamp embeddings was presented as a universal rule, which would prevent models with dedicated temporal architectures from using their native temporal processing. This was not the intended design. We have clarified this in three places: Section 2.2 now states that models with dedicated temporal architectures are expected to leverage them; Section 3.5 frames average-pooling as a reference protocol for models without temporal heads, not a universal requirement; and Section 3.6 requires submissions to declare which protocol was used, so results from temporal-head and reference-protocol models can be interpreted fairly. Our own baselines (Section 4.6) all use the reference protocol, which is now stated clearly in the paper.
>
> **C3**: The paired t-tests in Appendices D.1 and D.3 are appropriate there because each model serves as its own control: the same model is evaluated with and without a specific intervention (e.g., multi-spectral vs. RGB inputs) on identical data. This well-defined pairing structure is absent at the capability level in Figure 3, where we are comparing fundamentally different models across datasets with heterogeneous scales, tasks, and metrics. Constructing an artificial pairing in this setting would violate the assumptions of a paired test. Bootstrapping over the aggregated IQM scores is the statistically appropriate tool for quantifying uncertainty here. We have updated Section 3.4 and the Figure 3 caption to make explicit that models with overlapping confidence intervals should not be treated as definitively ranked, and to clarify why bootstrapped CIs are the appropriate uncertainty measure at the capability level.
>
> **C4**: We agree that assessing whether the Multi-Spectral-Dependent capability reflects a broadly representative signal is important. As noted, we already report per-dataset Cohen's d effect sizes in Appendix D.4, which make clear that `burn_scars` exhibits substantially larger sensitivity to band exclusion than other datasets. This concentration reflects the well-known physical reliance of burned area detection on specific spectral bands (e.g., NIR/SWIR), rather than an artifact of the evaluation protocol. To further assess robustness, we computed the capability ranking under a leave-one-out procedure, recomputing the ranking after excluding each multi-spectral dataset in turn (Appendix D.9). Removing `burn_scars` does not materially alter the relative ordering: the top-ranked and bottom-ranked models are unchanged across all leave-one-out conditions, and most models shift by at most one or two positions. Some model-specific sensitivities are observed (e.g., Satlas-Swin-100M shifts substantially upon exclusion of CloudSen12), but these reflect model-dataset dependencies rather than single-dataset dominance of the overall ranking.
>
> **C5**: Building on the leave-one-out analysis above, we applied the same procedure to the Multi-Temporal capability, recomputing the ranking after excluding each temporal dataset in turn (Appendix D.9). The ranking is highly stable: the top-ranked and bottom-ranked models remain unchanged across all leave-one-out conditions, and most models shift by at most one or two positions. Larger rank changes for specific models when PASTIS-R is excluded reflect model-specific dependencies rather than dominance of a single benchmark. This supports that the Multi-Temporal capability captures a consistent temporal performance signal across datasets.
>
> **C6**: We have added a rough estimate of GPU hours in Appendix Section C.
>
> **C7 and C8**: We note the reviewer's comments and plan to address them in due time.

---

> ### Author Response · Authors · 2026-04-22
> **Response to Reviewer - Editorial**
>
> **E1**: We have restructured the Introduction into a separate Related Work section.
>
> **E2**: All citation issues have been corrected: Faster R-CNN now cites Ren et al. (2015, NeurIPS); seven duplicate references (Ahlswede et al. 2022, Bountos et al. 2024, Clasen et al. 2024, Gourmelon et al. 2022, Kerner et al. 2025, Nascetti et al. 2023, Toker et al. 2022) consolidated to single canonical keys. Additional corrections: PhilEO Bench updated to IGARSS 2024; Hoffmann et al. (Chinchilla) title and venue corrected; Corley et al. (2024) moved to CVPR Workshops (PBVS); Gourmelon et al. and Ahlswede et al. updated to final published ESSD versions.
>
> **E3**: We added a note at the start of Section 3 clarifying that dataset splits, metrics, and IQM aggregation are framework-agnostic. TerraTorch tooling is offered for model convenience but is not required for leaderboard participation.

---

### Review · Reviewer_sVJZ · 2026-04-28

**Summary Of Contributions:**

The paper introduces Anonymized-Bench, a benchmark EO datasets collection, in an effort to standardize evaluation approaches across different EO tasks and models. Anonymized-Bench contains 19 different pre-existing publicly available datasets, such as TreeSatAI and NASA Burn Scars. Datasets were selected on the basis of how challenging they are, licensing permissiveness and to ensure diversity across the benchmark. Model performance across the benchmark is analyzed according to three “capability groups”. Task-centric capability relates to the task the model is being asked to perform, such as classification versus per-pixel outputs in segmentation. Input-centric capability relates to the input’s characteristics, such as the modality and resolution. Lastly, there is Core Capability, which is intended to convey model performance across all tasks. Anonymized-bench also includes an evaluation protocol with guidance on how models should be fine-tuned to the task and subsequently evaluated. The hyperparameter optimization is fairly constrained— search spaces must apply to at least 4 datasets, with each distinct search space documented and justified, and search budget is limited to at most 16 trials per dataset. Pre-processing and augmentation are required to be identical across all datasets for any given model, as well. Performance is then aggregated and results are discussed.

**Additional Comments:**

- What is “core”? Is it just an aggregation of models’ scores across the other datasets/tasks, or somehow a new dataset containing a subset of examples in most of the other datasets? According to table 3, why are not all datasets included in core? I would appreciate some more elaboration on how core is made and the rationale for design choices behind it.
- Section 3.7: Is there a reason for choosing the specific reference models that the authors selected?
- I believe there is an error in Figure 1: DinoV3-ViT-L Sat is labeled as an EO specialized model in Figure 1, but authors state it is trained on RGB images in Table 4 and Section 4.2.

**Audience:**

Yes

**Audience Explanation:**

Not a lot of work has focused on comparing performance across many different classes of EO models on different tasks; as such the results are novel and interesting (especially Sections 4.3-4.5, and ablations on frozen vs fine-tuning and on temporal processing).

**Broader Impact Concerns:**

None—well addressed by the broader impact statement.

**Claims And Evidence:**

Yes

**Claims Explanation:**

While I vote yes here, there are caveats behind it: the claims made are indeed supported by the evidence that is available in the paper, but the statistical significance of this evidence could be more firmly established. Otherwise put, the authors make claims in the paper that are broader than the experiments properly indicate; I believe that these claims should be softened.

Because EO is a large and diverse domain, limiting the evaluation to only 11 unique model families (e.g. considering two sizes of ResNet as one unique model family) provides narrow coverage. For instance, the set of models analyzed is light on the inclusion of SOTA weather FMs, despite the impressive performance of these models as demonstrated by other papers. Furthermore, the evaluated foundation models are heavily biased toward ViT and CNN architectures, with no representation of diffusion- or graph-based FMs. Accordingly, the following are examples of claims that generalize to *all* EO models (or all EO models with a given architecture/other characteristic), despite the limited representation within experiments:
- Section 4.5, “Architectural inductive biases interact strongly with task formulation”: The experiments feature only 11 unique model families to represent 4 architectures (ResNet/ConvNext/Swin/ViT) for just 3 task types. On an intuitive level I agree that this claim is likely true, but maintain the possibility that it is not necessarily true—the results in the paper are not broad-reaching enough to fully conclude one way or the other.
- Section 4.2, “EO-specialized models such as TerraMind and Prithvi consistently lead multi-spectral and multi-temporal capabilities, while natural-image pretrained models such as DINOv3-ViT-L-SAT and ConvNeXt variants excel in high-resolution RGB/NIR settings”
- Section 5, “EO has not exhibited the scaling laws observed in LLMs or image generation”: scaling laws do seem to apply for a given model on a given EO task, though I agree we perhaps don’t see “emergent capabilities” or similar phenomena exhibited for models like LLMs. I would request that the authors soften or add more nuance to this claim as well.
Authors do state in the limitations section 5.1 that they do not evaluate an exhaustive set of EO models, but it would be helpful to include this softening more clearly throughout Section 4.

Additionally, the renormalization tactic for aggregation across tasks (Section 3.7), while better than rank-based aggregation, still seems a little bit limited and results could be highly influenced by the choice of reference model set. A more robust strategy such as performance profiles could be helpful in this situation ([https://tmigot.github.io/posts/2024/06/teaching/](https://tmigot.github.io/posts/2024/06/teaching/)).

**Requested Changes:**

- The authors should include an appendix showing results across different aggregation strategies such as renormalization vs performance profiles, to show how robust the results are to choice of aggregation.
- The authors should soften claims such as those in Sections 4 and 5 detailed above.

---

> ### Author Response · Authors · 2026-05-13
> **Response to Reviewer sVJZ**
>
> We thank the reviewer for the positive vote and the constructive suggestions. Below we address each of the reviewer's points.
>
> ### C1: Limited coverage of EO model families (weather FMs and others)
>
> We agree that broader coverage would strengthen the benchmark, and we have added a caveat in Section 4.1 stating that our model selection, while representative of widely used GeoFMs, is not exhaustive. Weather foundation models such as Aurora and GenCast have shown impressive results in their domain, but their available checkpoints are not transferable to the discriminative downstream tasks covered by our benchmark. Similar considerations apply to domain-specific FMs in adjacent fields such as flood modelling. We believe our selection covers a substantial portion of commonly used remote-sensing models, while acknowledging that future benchmark iterations should expand this coverage as more general-purpose checkpoints become available.
>
> ### C2: Architectural bias toward ViT/CNN; no diffusion or graph FMs
>
> We agree that graph neural networks are not represented in our evaluation. Regarding diffusion models, we note that their underlying architectures are typically ViT or CNN based; the difference lies in the training objective rather than the architecture itself. TerraMind is in fact pretrained with a generative diffusion objective. Because Anonymized-Bench targets discriminative downstream tasks, we have not included diffusion-based generative models adapted for discriminative use, but we agree this is a meaningful direction for future extensions.
>
> ### C3: Section 4.5 claim about inductive biases and task formulation
>
> We have rephrased this claim to be more precise: "Within the evaluated model families and tasks, inductive biases appear to interact with task formulation."
>
> ### C4: Section 4.2 claim about EO-specialized vs natural-image pretraining
>
> We have added an explicit caveat at the end of Section 4.1 noting that our model selection is not exhaustive and that conclusions should be contextualized accordingly. Thus, the trend reported in Section 4.2 holds within our evaluated set and is not intended as a universal statement about all EO-specialized or natural-image pretrained models.
>
> ### C5: Section 5 claim about scaling laws in EO
>
> We have softened this claim to: "While our analysis confirms that larger models tend to perform better, we do not observe broad, cross-task scaling behavior or emergent effects of the kind reported for LLMs or image generation."
>
> ### C6: Aggregation via renormalization (Section 3.7)
>
> We have computed performance profiles (Dolan and Moré, 2002) on our main fully-finetuned results, which do not depend on a single reference model set and are insensitive to score scaling. The figure and description are included in Appendix B, and we now point to this analysis in Section 3.7: "In addition, we have also included performance profiles (Dolan and Moré, 2002) in Appendix B which do not rely on a single reference model." The profiles are broadly consistent with our renormalization-based ranking, supporting the robustness of our conclusions to the aggregation strategy.
>
> ### C7: What is "Core"?
>
> Core is an aggregation capability defined over a balanced subset of the benchmark's datasets, not a new dataset and not an aggregation over all 19 datasets. It serves as a computationally efficient yet discriminative entry point for users who cannot afford the full benchmark. The selection criterion, now expanded in Section 2.2, applies the same discriminative-power criterion used in Section 2.1, with the constraint that no single task type, resolution range, or spectral profile is over-represented. Thus, not all datasets appear in Core in Table 3: including all would defeat the purpose of a compact, balanced subset and reintroduce the imbalances Core is designed to mitigate.
>
> ### C8: Section 3.7 reference model selection
>
> The reference set consists of all 14 models reported in Table 4, i.e. all models evaluated on every dataset in our main experiments. It spans architectural families (ResNet, ConvNeXt, Swin, ViT), parameter scales (25M to 600M), pretraining domains (natural-image and EO-specific), and the observed performance range, so that the $[0, 1]$ bounds reflect a representative envelope of model behavior. We have added this description to Section 3.7. As an additional safeguard, the reference-free performance profile analysis added in Appendix B yields a broadly consistent ranking, indicating that our conclusions are not driven by the specific reference set.
>
> ### C9: DINOv3-ViT-L-SAT label in Figure 1
>
> The reviewer is correct that the label was inconsistent with the description in Section 4.2. DINOv3-ViT-L-SAT derives from standard DINOv3-ViT-L distilled on the proprietary SAT-493M dataset: 493 million high-resolution RGB satellite images at 0.4 to 0.6 m resolution. While it is single-modality, we consider it EO-specific because its pretraining corpus is entirely satellite imagery.

---

### Review · Reviewer_ZRpc · 2026-05-05

**Summary Of Contributions:**

The paper introduces **Anonymized-Bench**, a benchmark for Geospatial Foundation Models (GeoFMs) covering 19 permissively-licensed EO datasets across classification, segmentation, regression, object detection, and instance segmentation. Main contributions:

1. **Benchmark suite with capability groups**: 19 permissively-licensed EO datasets spanning multiple modalities in TACO format, organized into multiple overlapping capability axes that surface task/modality-specific strengths instead of a single aggregate ranking.
2. **Prescriptive evaluation protocol with aggregation**: standardized experimentation (5 repeat seeds, augmentations, prescribed decoders and explicit rules for SAR and multi-modal inputs), supporting ablations (e.g., freezing the encoder reorders >20% of model pairs), with sufficient metric
3. **Empirical study of 14 models with public tooling** Study across multiple models and find that no single model dominates: natural-image FMs lead high-res RGB/NIR; EO-specific FMs lead multi-spectral and multi-temporal; Clay-V1 ViT-B tops Core. Also this contain public accessibility, which have TerraTorch integration and a public HuggingFace leaderboard.

**Audience:**

Yes

**Audience Explanation:**

GeoFM benchmarking is an active area and the question of whether EO-specific FMs beat natural-image FMs is worth investigating. This paper contributes a methodological template (capability decomposition, prescriptive protocol, IQM aggregation) and a empirical answer of how to make the choice are useful. The work also make this benchmark more accessible by setting up leaderboard, and have terratorch integration, etc.

**Broader Impact Concerns:**

One concern is geographic imbalance toward Europe/North America may yield over-optimistic estimates in underrepresented regions — and recommends independent regional validation if possible.

No ethical concerns warrant gating the paper.

**Claims And Evidence:**

No

**Claims Explanation:**

I am happy with most aspects of this work — the experimental setup, statistical aggregation, and ablations are well executed. My concern is one of the main claim in the paper: **"no single model dominates"** across the nine capabilities, but this conclusion is drawn over a model pool that excludes multimodal LLMs (MLLMs) entirely. Given that 2026-era MLLMs (frontier closed-source GPT-5.X / Gemini, open-source Qwen-VL, and EO-specialized variants like GeoChat / EarthGPT / RemoteCLIP) have shown strong zero-shot and few-shot capability on RS classification and detection, it is genuinely possible that a single MLLM does dominate a meaningful subset of capabilities. Without evaluating at least one or two MLLMs, the "no single model dominates" claim is weaken, and the abstract's framing as a benchmark for "general-purpose geospatial intelligence" might be outdated in 2026. This is why I lean weakly toward "No" rather than "Yes" here.

**Requested Changes:**

**High priority:**

1. **Include multimodal LLMs (MLLMs) as baselines or explicitly justify their omission.** As of 2026, frontier MLLMs — closed-source such as GPT-5.X and Gemini series, open-source such as Qwen2.5-VL and InternVL3 — and EO-specific MLLMs like GeoChat / EarthGPT / RemoteCLIP are highly capable on classification and detection tasks via zero-shot or few-shot prompting. Excluding them entirely is a major gap for a benchmark that aims to be the community standard for "general-purpose geospatial intelligence", and directly weakens the headline claim that no single model dominates.
At minimum, the paper should: (a) evaluate 2–3 strong general-purpose MLLMs (open and closed-source) and at least one EO-specialized MLLM on the Classification and Detection capabilities, and (b) extend the protocol §4 to accommodate prompting-based evaluation paradigms (zero-shot / few-shot / in-context) alongside fine-tuning. If a full evaluation is out of scope (e.g., due to compute or modality mismatch with multi-spectral/SAR), this exclusion should be explicitly motivated and surfaced as a limitation. Otherwise, we need a clear reason for not doing this

**Minors**
1. in §3.1 it filter out EuroSAT, Sen1Floods11 because they are saturated, I wonder if all model are saturated in those datasets? If we introduce them, will this lead to a change in ranking?

---

> ### Author Response · Authors · 2026-05-13
> **Response to Reviewer ZRpc**
>
> We thank the reviewer for the positive overall assessment and for the constructive framing of the MLLM concern. We address both points below.
>
> ### Inclusion of multimodal LLMs (MLLMs)
>
> We agree that MLLMs are an important class of models for the broader EO landscape, and that their absence warrants explicit motivation. In the revised manuscript we have addressed this at four points throughout the paper, rather than as a single isolated note:
>
> 1. Introduction: we now explicitly position MLLMs in the GeoFM landscape and cite recent evidence (GPT-5, Gemini, GEOBench-VLM) showing that frontier MLLMs continue to struggle with complex spatial and geospatial reasoning tasks. We argue that vision GeoFMs remain a strong and practical choice for EO workloads due to their domain-specific pretraining and substantially smaller model size (often 10 to 100x fewer parameters).
>
> 2. Related Work: we have added a dedicated paragraph discussing GEOBench-VLM, which provides the systematic MLLM evaluation on EO tasks. That work spans 31 fine-grained tasks designed for VLM evaluation and reports that state-of-the-art VLMs achieve limited accuracy on geospatial tasks, with consistent failure modes on small-object reasoning, spatial precision, and temporal change understanding. We view Anonymized-Bench and GEOBench-VLM as complementary efforts targeting different model paradigms.
>
> 3. Model selection (Section 4.1): we have added an explicit caveat that our selection, while representative of the most commonly used models in the remote sensing community, is not exhaustive and that conclusions should be contextualized accordingly.
>
> 4. Limitations (Section 5.1): we have added a paragraph explicitly motivating the MLLM exclusion. Anonymized-Bench targets the GeoFM paradigm, namely encoder-style models adapted to downstream tasks via full fine-tuning. The prescriptive protocol (HPO, decoders, 5-seed repeats, IQM aggregation) is defined around this paradigm. MLLMs are predominantly RGB-only, do not natively consume the multi-spectral (Sentinel-2, EnMAP, Landsat) or SAR (Sentinel-1) inputs central to most of our 19 datasets, and cannot produce the dense pixel-level outputs required by segmentation and regression tasks. At best they could be evaluated on a small fraction of our capability groups, which would not provide a comparable signal to the full benchmark. We frame integration of prompting-based evaluation paradigms (zero-shot, few-shot, in-context) into capability-driven GeoFM benchmarking as an explicit direction for future work.
>
> Finally, we have softened the abstract's closing sentence from "general-purpose geospatial intelligence" to "general-purpose Geospatial Foundation Models" to align the framing with the actual scope of the benchmark.
>
> ### Minor: Saturated datasets (EuroSAT, Sen1Floods11)
>
> The reviewer asks whether all models are saturated on these datasets and whether their inclusion would change rankings. This is exactly the criterion underlying our selection process (Section 2.1, "Challenging and Discriminative"): we discarded datasets where inter-model differences fell below the variability across random seeds. EuroSAT and Sen1Floods11 are well-documented saturated benchmarks. We have strengthened Section 2.1 to make this criterion more explicit and added citations to GEO-Bench v1, which already documented EuroSAT saturation, and to three recent GeoFM evaluations (Prithvi-EO-V2, TerraMind, Clay) that consistently confirm this pattern. Including these datasets would not meaningfully change rankings because the seed-level variability dominates the inter-model spread, which is the formal definition of saturation we used.
>
> ### Broader Impact
>
> We thank the reviewer for confirming the broader impact statement is appropriate. We agree with the recommendation for independent regional validation in underrepresented regions, which is already noted in our broader impact statement.

---

> ### Comment · Reviewer_ZRpc · 2026-05-13
>
> Thanks for the revision. The added MLLM positioning and softened abstract address the scope concern. My remaining issue is with how the cited MLLM evidence is *characterized*.
>
> **The cited numbers don't support the "struggle" / "limited accuracy" framing.** Direct comparison on EuroSAT (which the paper excluded as saturated):
>
> | Model | EuroSAT acc. |
> |---|---|
> | Fine-tuned encoders (per paper) | ~98% |
> | Gemini 2.5 zero-shot RGB (Mallya) | 66.3% |
> | Gemini 2.5 zero-shot multi-spectral (Mallya) | 69.1% |
> | CLIP zero-shot | ~60% |
>
> Plus GPT-5 scores 41–84% on 8 spatial benchmarks (Cai Table 3), well above 25–34% random baselines. A more accurate framing would be **"MLLMs lag fine-tuned specialists by ~30 points on EO classification"** — the gap supports the scoping decision, the word "struggle" overstates it, which sounds like 20% accuracy or something
>
> **Mallya itself contradicts the §5.1 multi-spectral claim.** §5.1 asserts MLLMs "do not natively consume multi-spectral or SAR inputs," but Mallya paper shows Gemini 2.5 reaches 69.1% on multi-spectral EuroSAT with a simple adaptation. The structural argument holds for *vanilla* MLLMs only.
>
> **My pending requests:**
> 1. **Hedge the language** — e.g., "MLLMs lag fine-tuned specialists by ~X points despite non-trivial zero-shot accuracy" — and acknowledge Mallya shows MLLM multi-spectral adaptation is feasible.
> 2. **Some zero-shot MLLM data point** for example, on the Classification capability (e.g., Gemini 3 / qwen3-vl on BigEarthNet V2 or TreeSatAI). The structural arguments don't apply here and this should not be hard to set up.
>
> Saturated-datasets minor: resolved.

---

> > ### Author Response · Authors · 2026-05-19
> > **Response to Reviewer ZRpc**
> >
> > We thank the reviewer for their insightful comments.
> >
> > **Request 1**: We have hedged the language to reflect the findings of Mallya more accurately in the introduction section and limitation section.
> >
> > **Request 2**: As suggested by the reviewer, we provide some data points for MLLMs. We choose two openly available MLLMs — Qwen2.5-VL-7B-Instruct and InternVL3-8B in a zero-shot setting on So2Sat and BigEarthNet-v2. Both models perform near chance level on So2Sat (accuracy ≈10–13%, vs. ~6% random for 17 classes) and achieve limited F1 on BigEarthNet-v2. We note that this zero-shot evaluation represents a lower bound on MLLM capability for these tasks: achieving competitive performance would likely require careful prompt engineering, class-description tuning, and strategies for handling large and imbalanced label spaces. A thorough and fair integration of MLLMs into the benchmark is an interesting and important direction for the field.
> >
> > | Dataset | Model | Accuracy | Macro F1 | Micro F1 |
> > |---|---|---:|---:|---:|
> > | so2sat | OpenGVLab/InternVL3-8B | 0.1005 | 0.0691 | — |
> > | so2sat | Qwen/Qwen2.5-VL-7B-Instruct | 0.1255 | 0.0915 | — |
> > | benv2 | OpenGVLab/InternVL3-8B | — | 0.2142 | 0.3282 |
> > | benv2 | Qwen/Qwen2.5-VL-7B-Instruct | — | 0.2022 | 0.2868 |

---

> > > ### Comment · Reviewer_ZRpc · 2026-05-20
> > >
> > > Both requests are addressed. The language is hedged, and the new zero-shot results (So2Sat 10–13% accuracy vs. ~6% random; BigEarthNet-v2 micro F1 0.29–0.33) give the benchmark a MLLM data point rather than relying on cited numbers alone.
> > >
> > > One caveat for the record: the test uses 7–8B open models on multi-spectral data, which seems a hard one for MLLMs, rather than the frontier models (GPT-5, Gemini 3) discussed earlier. The results might not be a representative estimate, but the authors say as much.

---

### Decision · Action_Editor_rt5J · 2026-06-18

**Recommendation:** Accept as is

**Additional Comments:**

This paper tackles earth observation, where there has been strong progress given recent foundation model advances. The goal of the paper is to standardize benchmarking and evaluation for this setting. It builds a framework with several different task types (e.g., classification, segmentation, ...) and an evaluation protocol. It then produces an analysis of popular approaches.

Reviewers were broadly positive about the work. They found the motivation to be strong and agreed that the work is going to be quite useful going forward. The main limitations identified have to do with the total number of evaluations and settings explored here (there can always be additional ones here given how general the area is, but this first cut is good) along with the specific justifications behind some claims. The latter have been resolved through the discussion and via some additions. I believe there is a great deal of valuable work in the paper and so I recommend acceptance.

**Audience:**

Yes

**Audience Explanation:**

Yes. there is a lot of interest in geospatial foundation models and related tasks.

**Claims And Evidence:**

Yes

**Claims Explanation:**

Yes, there is solid methodological and empirical evidence for the claims in this work.